# Expanding PROTACtable genome universe of E3 ligases

Yuan Liu [1,2,3], Jingwen Yang[1,2,3], Tianlu Wang[4], Mei Luo[1,2], Yamei Chen [1,2,3], Chengxuan Chen[1,2,3], Ze'ev Ronai [5], Yubin Zhou [4,6], Eytan Ruppin [7] ✉ & Leng Han [1,2,3,6] ✉

Proteolysis-targeting chimera (PROTAC) and other targeted protein degradation (TPD) molecules that induce degradation by the ubiquitin-proteasome system (UPS) offer new opportunities to engage targets that remain challenging to be inhibited by conventional small molecules. One fundamental element in the degradation process is the E3 ligase. However, less than 2% amongst hundreds of E3 ligases in the human genome have been engaged in current studies in the TPD field, calling for the recruiting of additional ones to further enhance the therapeutic potential of TPD. To accelerate the development of PROTACs utilizing under-explored E3 ligases, we systematically characterize E3 ligases from seven different aspects, including chemical ligandability, expression patterns, protein-protein interactions (PPI), structure availability, functional essentiality, cellular location, and PPI interface by analyzing 30 large-scale data sets. Our analysis uncovers several E3 ligases as promising extant PROTACs. In total, combining confidence score, ligandability, expression pattern, and PPI, we identified 76 E3 ligases as PROTAC-interacting candidates. We develop a user-friendly and flexible web portal (https://hanlaboratory.com/E3Atlas/) aimed at assisting researchers to rapidly identify E3 ligases with promising TPD activities against specifically desired targets, facilitating the development of these therapies in cancer and beyond.

Targeted protein degradation (TPD) is actively pursued as an emerging therapeutic strategy that can target proteins that were previously considered undruggable[1–3]. PROTAC, a major focus of TPD, recruits and binds to both E3 ligase and protein of interest (POI) simultaneously, resulting in the formation of an E3-degrader-POI ternary complex to induces POI ubiquitination and subsequent degradation by the proteosome[4]. Given the enormous therapeutic potential and unique mechanism of PROTAC, the field attracts great interest from both academia and the pharmaceutical industry, with ≥250 being tested for degradation by PROTACs[5]. PROTACs are capable of targeting many proteins that are traditionally considered as 'undruggable' with traditional small-molecule-mediated pharmacology[1,4]. A recent comprehensive analysis utilized lessons learned from previous PROTACs and depicted each target with applicable aspects including ubiquitylation, turnover rate, small-molecule binder availability, and intracellular localization, identifying ~1000 protein targets that are highly 'PROTACtable'[6].

[1]Department of Biostatistics and Health Data Science, School of Medicine, Indiana University, Indianapolis, IN, USA. [2]Brown Center for Immunotherapy, School of Medicine, Indiana University, Indianapolis, IN, USA. [3]Center for Epigenetics and Disease Prevention, Institute of Biosciences and Technology, Texas A&M University, Houston, TX, USA. [4]Center for Translational Cancer Research, Institute of Biosciences and Technology, Texas A&M University, Houston, TX, USA. [5]Cancer Center, Sanford Burnham Prebys Medical Discovery Institute, La Jolla, CA 92037, USA. [6]Department of Translational Medical Sciences, College of Medicine, Texas A&M University, Houston, TX, USA. [7]Cancer Data Science Laboratory, Center for Cancer Research, National Cancer Institute (NCI), National Institutes of Health (NIH), Bethesda 20892 MD, USA. ✉e-mail: eytan.ruppin@nih.gov; lenghan@iu.edu

Despite the extensive therapeutic potential of PROTAC, among over 600 E3s in the human genome[3,7–9], only a few are currently utilized in PROTACs. The PROTAC field is still in its early stages, in which the possible application of this technology is being explored. Notably, key component in PROTAC technology is the E3 degrader, yet limited effort has been made to identify additional E3 ligases that can serve PROTAC technology. Given the importance of ligand availability, and subcellular localization among other factors determining PROTAC effectiveness, E3 ligases are expected to play a central role in its functionality. These numerous factors make the search for new E3 ligases challenging. Much of the initial efforts have been devoted to pairing *VHL* or *CRBN* with different target proteins in order to save resources and expedite progress. PROTACs in clinical phases are heavily relying on recruiting either *VHL* or *CRBN*[1]. New E3 ligases for PROTAC design are hence called for to alleviate concerns on explored E3 ligases and to expand PROTAC therapeutical potential[1] for a variety of reasons. First, new E3 ligases may reduce on-target toxicities of PROTACs[1,4,10,11]. The function of PROTAC relies on the expression of the recruiting E3 ligase in the corresponding cells, and low-level expression of the E3 ligase in undesired cells will minimize the activity of the PROTAC thereby reducing the side effects[1,4]. For example, DT2216, a PROTAC recruiting *VHL* and targeting *BCL-XL*, utilized the poor expression of *VHL* in platelets and successfully overcome the side effect in platelets caused by the original small molecule inhibitors[12,13]. Second, new E3 ligases could also circumvent the acquired drug resistance that caused by genomic changes at the E3 ligase loci[1,10], which have been reported to impair the efficacy of PROTACs. For example, in myeloma patients, genetic aberrations in CRBN, the most widely used *CRBN* recruiters[14], could evade *CRBN*-based PROTACs[1]. Thirdly, additional E3 ligases may target more challenging POIs. For example, large-scale degradation screenings on human kinome revealed that the degradation target space varied among E3 ligases[15,16]. This variation of different E3 ligases against specific targets may result from the unique PPIs between E3 ligases and targets[16] and hence, additional E3 ligases may extend PROTAC against more new targets.

Although comprehensive analyses have been conducted to depict PROTAC tractability of targets[6], the space of E3 ligases, the counterparts needed to design PROTACs to degrade these POI targets, have not yet been systematically quantitatively characterized. Well-understood E3 ligases may lead to site-specific and target-specific degradation, enabling precision targeted protein degradation. To address this knowledge gap in the TPD field, we here go markedly beyond[6], whose work has focused on the targets degraded by PROTAC. Here we have assembled a comprehensive list of E3 ligases (Fig. 1a) and leveraged large-scale data resources, experimental evidence, and artificial intelligence (AI) tools to delineate seven key different dimensions of E3 ligases: (1) chemical ligandability that characterizes the availability of binder(s) for the E3 ligase; (2) expression patterns of E3 ligases in tumor and normal samples at both bulk and single-cell levels; (3) PPI of E3 ligases that further characterize their potential protein targets; and (4) other important factors that should preferably be considered in the PROTAC design, including structure availability, functional essentiality, cellular localization, and PPI interfaces of E3 ligases (Fig. 1b). Current data resources may reflect known biases. Thus, we've incorporated options for users to select data sources based on their preferences. In addition to searching for new E3 ligases for PROTAC, this information of E3 ligases can also be leveraged by other TPD fields, such as proteolysis-targeting antibody (PROTAB) and molecular glues, which all rely on E3 ligases and have the need to expand and diversify PROTACtable E3 ligases[17,18]. Furthermore, even though cancer therapeutics account for most of the newly developed PROTACs[1], our comprehensive analysis of E3 ligases could also be migrated beyond cancer to therapeutic development in other fields, such as for treating neurological and/or immunological disorders.

## Results

### Constructing a comprehensive collection of E3 ligases

Over 600 E3 ligases were reported to be encoded by the human genome[1]. To achieve comprehensive coverage of E3 ligases for PRO-TAC, we generated a collective E3 ligase set by combining several credible diverse human-curated E3 ligase lists, including Ge et al.[9], UbiHub[8], and UbiBrowser2.0[7]. Ge et al. assembled an E3 ligase list comprising 882 genes from the Ubiquitin and Ubiquitin-like Conjugation Database[19] that performed keyword-based literature parsing and Hidden Markov Model prediction; UbiHub constructed an E3 ligase list consisting of 670 genes mainly through signature domains searching[8]; and UbiBrowser2.0 collected E3-substrate interaction (ESI) that involves 404 genes by parsing literature[7]. E3 ligases that have more comprehensive evidence to be involved in the UPS system may be more likely to be co-opted in PROTAC[1,20]. We therefore assigned a confidence score (1–6; 6 is the best) for E3 ligase from each source to indicate the confidence level (Fig. 1a, Supplementary Fig. 1; Supplementary Data 1). A higher score indicates more available information on a specific E3 ligase in terms of function and substrate.

A total of 1075 unique E3 ligases were collected from the above three credible sources, however, only 12 (1.1%) E3 ligases were co-opted in PROTAC design so far. Reassuringly, most of these 12 E3 ligases co-opted in PROTACs were assigned a high score of 5 or 6, including *VHL*, *CRBN*, and *MDM2* (Fig. 1a). The two E3 ligases advanced to clinical trials, *VHL* (NCT04886622) and *CRBN* (NCT03888612, NCT04072952, and NCT05080842), were both scored at 6. Experimentally explored E3 ligases, such as *KEAP1*[21], *MDM2*[22], and *DCAF16*[23], were scored at either 5 or 6. A score of 5 or 6 also indicated these E3 ligases appeared and were cross-validated in three E3 ligase lists. This aligns well with the current progress of PROTAC development that E3 ligases were co-opted only when sufficient a priori knowledge has existed[1]. Notably, beyond these co-opted E3 ligases, 275 (25.6%) E3 ligases had obtained with scores of 5 or 6. For example, *RNF4*, a co-opted E3 ligase, has well-documented roles in UPS and 12 known E3-substrate interactions (ESIs). Most of the E3 ligases listed above have not yet been co-opted in PROTACs but may serve as potentially novel co-opted E3 ligases, such as HECT, UBA And WWE Domain 1 (*HUWE1*) and F-Box Protein 7 (*FBXO7*). *HUWE1* endogenously promotes the degradation of MCL1 Apoptosis Regulator, BCL2 Family Member (*MCL1*), and *FBXO7* regulates the ubiquitination of Mitofusin 1 (*MFN1*)[8]. These E3 ligases have scored similarly to the co-opted E3 ligases and may be prioritized in the search for new E3 ligases in PROTAC development.

### Depiction of the ligandability of E3 ligases

A critical and necessary step to design a potent PROTAC is to identify an appropriate ligand to bind E3 ligase[24]. To leverage the knowledge of existing ligands, we systematically collected 3 categories of experimental records including drugs (from DrugBank[25] and DGIdb[26]), small-molecule ligands (from ChEMBL[27]), and electrophiles (a.k.a. covalent binder, from Streamlined Cysteine Activity-Based Protein Profiling, SLCABPP[28]).

We identified 686 (63.8%) E3 ligases that interact with known ligands from at least one category of ligand sources (Fig. 2a, Supplementary Fig. 2a). Specifically, 127 (11.8%) E3 ligases had evidence of targeting by or interacting with drugs in DrugBank[25] and/or DGIdb[26], 145 (13.5%) E3 ligases have bioactive ligands in ChEMBL, and 626 (58.2%) can interact with electrophiles in SLCABPP (Fig. 2a, Supplementary Fig. 2b). Inspired by Pharos, a druggable genome resource[29], we labeled E3 ligases into E3drug, E3chem, E3cova, and E3dark (see "Methods"). Among the identified E3 ligases, 127 (11.8%) were labeled as E3drug, 75 (7.0%) as E3chem, and 484 (45.0%) as E3cova. In addition, we quantified the number of ligand categories associated with the E3 ligases. We identified that 55 (5.2%) E3 ligases can interact with all three sources of ligands (Fig. 2a, Supplementary Fig. 2a), including the clinically used *VHL* and *CRBN*, and the experimentally explored *KEAP1*, *XIAP*, *MDM2*, *BIRC2*, and *AHR*. Beyond these E3 ligases, 48 have not yet

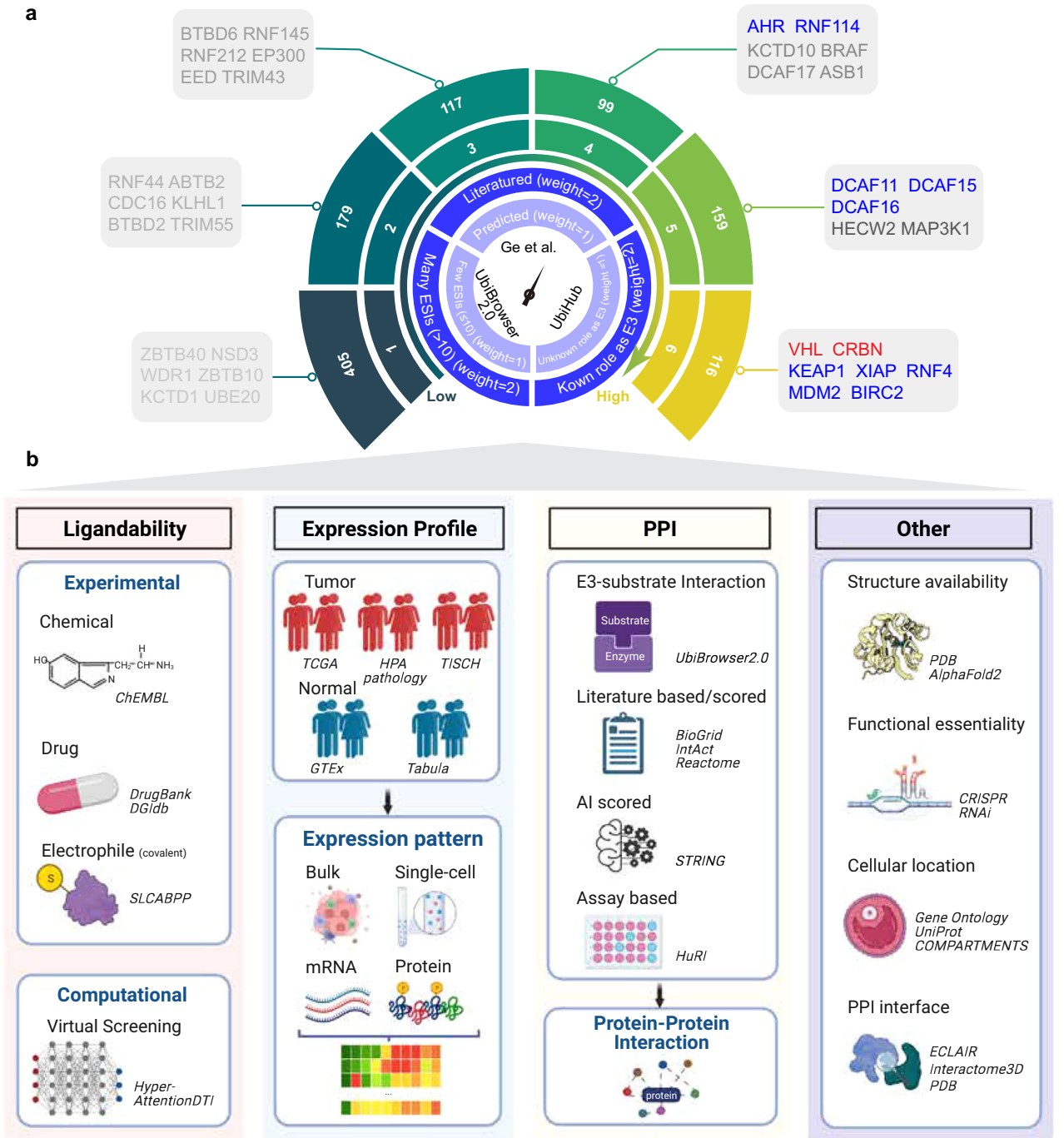

**Fig. 1 | Schematic workflow of collecting and annotating E3 ligases. a** Illustration of collecting and scoring E3 ligases in inner circle, while the distribution of scores of E3 ligases and co-opted E3 ligases in outer circle. E3 ligases in red were proceeded to PROTAC clinical trials. E3 ligases in blue were explored in PROTAC experiment. **b** We comprehensively characterized E3 ligases chemical ligandability, expression patterns, protein–protein interactions (PPI), structure availability, functional essentiality, cellular location, and PPI interface. Created with BioRender.com.

been reported to be co-opted in PROTAC, pointing them as potential candidates for expanding E3 ligases for PROTACs. For instance, Tripartite-motif protein 24 (*TRIM24*)[30] has evidence of available ligands from drug, ChEMBL, and SLCABPP (Fig. 2b), example ligands including salicyladehyde (CHEMBL108925) and the inhibitor (DGIdb: 252166607). In addition to the count of ligand sources, we also quantified the total number of ligands per E3 ligase. 77 (7.2%) E3 ligases have over 300 ligands, including 7 out of 12 co-opted E3 ligases (Fig. 2c). Reassuringly, the two E3 ligases used in clinical, *VHL* and *CRBN*, have a

very high number of interacting ligands reported (termed ligandability), having 2064 and 2722 ligands, respectively (Fig. 2d). Other co-opted E3 ligases also have a high number of ligands, such as *BIRC2* (1328) and *AHR* (786) (Fig. 2d). Notably, beyond these co-opted E3 ligases, a potentially novel E3 ligase, Histone Deacetylase 6 (*HDAC6*), has over 10,000 identified ligands more than each of the other E3 ligases we have explored, and 70 (6.5%) E3 ligases possessed over 300 ligands, such as Poly(ADP-Ribose) Polymerase 1 (*PARP1*) and Interleukin 1 Receptor Associated Kinase 4 (*IRAK4*).

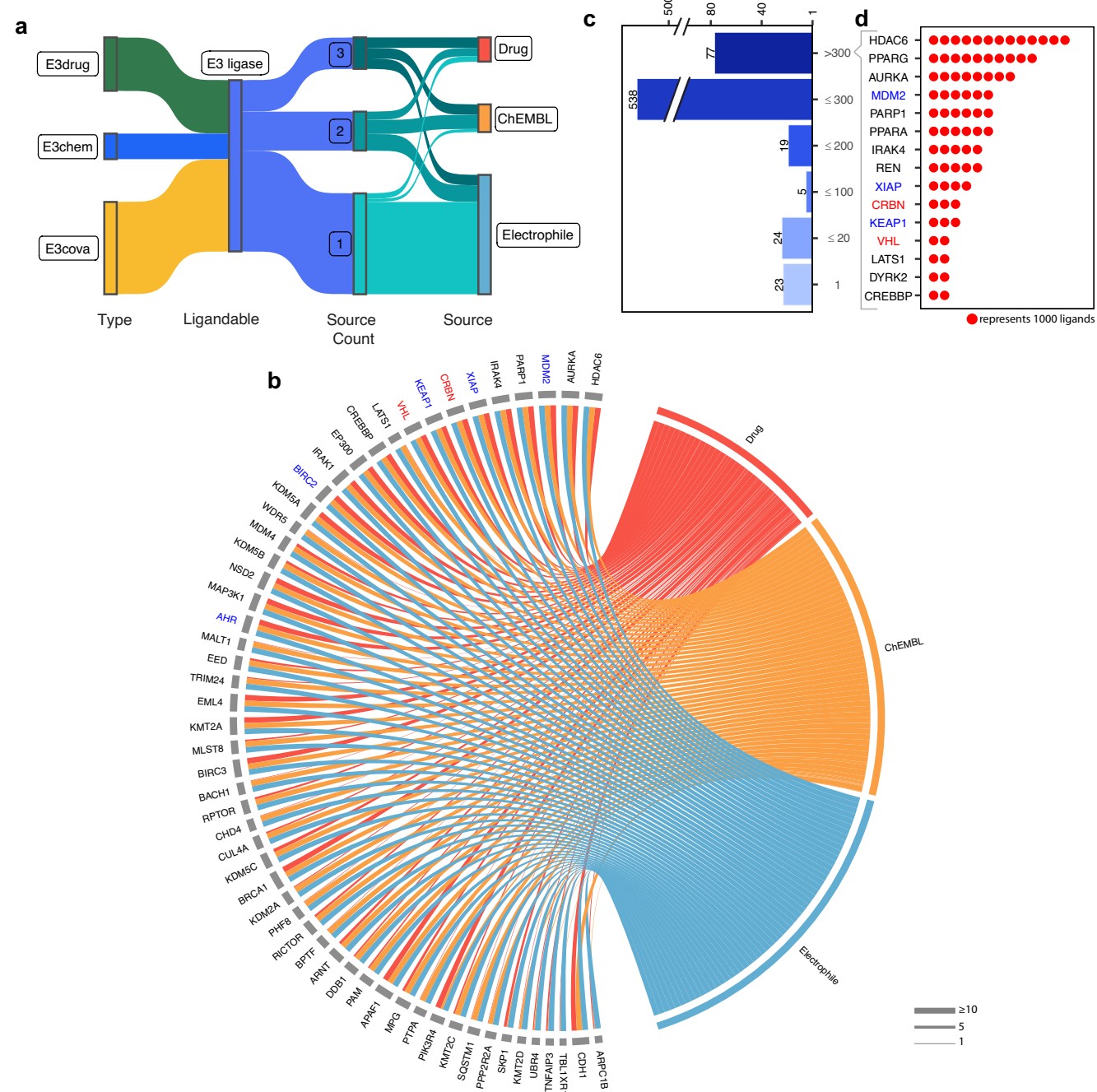

Fig. 2 | Depiction of the ligandability of E3 ligases. a Distribution of the number of categories of ligands identified for each E3 ligase from multiple data sources. b Chord chart for E3 ligases with 3 types of ligands. c Circular barplot representing the distribution of unique ligands of E3 ligases. d E3 ligases with top number of unique ligands and one dot representing 1000 ligands. E3 ligases in red were proceeded to PROTAC clinical trials. E3 ligases in blue were explored in PROTAC experiment.

To search for new possible drugs that may facilitate the recruitment of E3 ligases, we adopted a deep-learning-based virtual screening ligand model, HyperAttentionDTI[31], to search for potential interactions between E3 ligases and drugs in DrugBank[25]. The model captured complex interactions between atoms and amino acids through an innovative attention mechanism and achieved improved performance over the state-of-the-art baselines, and the new drug-target interactions falls into top virtual screening predicted interactions in the test case study. After evaluating and applying this model (Supplementary Fig. 3), we obtained predicted drug-target interactions, including for potentially novel E3 ligases, such as Makorin Ring Finger Protein 1 (MKRN1), having PPIs with key tumor drivers, *TP53* and *APC32*, and

predicted to interact with DB13955 (Estradiol dienanthate) and DB03017 (Lauric acid). Utilizing these drugs may increase the ligandability of potentially novel E3 ligases that target additional tumor targets. Taken together, our systematic analysis exploited five large-scale data sources and revealed a number of new E3 ligases with tens of available ligands that may be utilized for PROTAC development.

**The expression landscape of E3 ligases in tumors**
The function of PROTAC relies on the expression of the recruiting E3 ligase in the corresponding cells and thus the expression level of an E3 ligase in targeted and undesired cells influences the efficacy and side effects of PROTAC[4]. Charting the expression profiles of E3 ligases is

critical for enabling precision TPD and has attracted the interests of both academia and industry[1]. We characterized the expression landscape of each E3 ligase by exploring transcriptomics and proteomics datasets.

We first evaluated the expression level of E3 ligases across 33 cancer types in The Cancer Genome Atlas (TCGA)[32–34], and we identified a total of 765 (71.2%) E3 ligases highly expressed [log$^{(TPM+1)}$ > 4] in at least one cancer type in TCGA. The number of highly expressed E3 ligases in each cancer type ranged from 216 in Liver hepatocellular carcinoma (LIHC) to 529 in Acute Myeloid Leukemia (LAML) (Fig. 3a). Our analysis revealed that all existing co-opted E3 ligases were highly expressed in at least one cancer type, and most are ubiquitously highly expressed (Supplementary Fig. 4a). Among them, 6 co-opted E3 ligases, such as *MDM2* and *KEAP1*, were highly expressed in over 30 cancer types, suggesting their high potential for broad usage in multiple cancer types (Supplementary Fig. 4a). Beyond these co-opted E3 ligases, we identified 363 (33.8%) potentially novel E3 ligases highly expressed in over 30 cancer types, such as *HUWE1* and *FBXO7*, indicating the high potential of these E3 ligases for pan-cancer usage (Fig. 3a).

Since protein expression level is a direct proxy to measure protein acitivity[35], we further evaluated the protein expression level of E3 ligases in tumors from human protein atlas pathology (HPA pathology)[36]. The number of highly expressed (≥20% of samples highly expressed[37]) E3 ligases in each cancer type ranged from 380 in lymphoma to 617 in thyroid cancer (Fig. 3b). We identified 780 (72.6%) E3 ligases highly expressed in at least one tumor type at the proteomics level. Four co-opted E3 ligases, such as *CRBN* and *MDM2*, were highly expressed in over 15 cancer types (Supplementary Fig. 4b). Beyond those co-opted E3 ligases, 371 potentially novel E3 ligases were highly expressed in over 15 cancer types. For example, both *FBXW7* and *TRIM28* were highly expressed in all cancer types (Fig. 3b).

Integration of both proteomic and transcriptomic data demonstrated the high concordance of these highly expressed E3 ligases in tumors. For example, *CRBN* is highly expressed in the prostate tumor at both transcriptomic and proteomic levels, which aligned with the usage of *CRBN* hijacked by a PROTAC targeting androgen receptor (*AR*) in clinical trials in prostate cancer (NCT03888612; NCT04428788)[1]. Another example is that *CRBN* is highly expressed in breast invasive carcinoma (BRCA) in TCGA and breast cancer in HPA pathology, which aligned with in-trial PROTACs targeting estrogen receptor (*ER*) in breast cancer (NCT04072952; NCT05080842)[1]. We also identified potentially novel E3 ligases highly expressed in tumors at both transcriptomic and proteomic levels. For example, Nitric Oxide Synthase Interacting Protein (*NOSIP*) was highly expressed in pancreatic adenocarcinoma (PAAD) in TCGA and was also highly expressed in 10 of 11 pancreatic cancer samples in HPA pathology (Fig. 3a, b). Similar to *NOSIP* in pancreatic cancer, Ring Finger Protein 130 (*RNF130*) was highly expressed in breast invasive carcinoma (BRCA) in TCGA and was also highly expressed in 11 of 12 breast cancer samples in HPA pathology. 42 E3 ligases, consisting of 2 co-opted E3 ligases, *MDM2* and *DCAF11*, and 40 potentially novel E3 ligases, such as *TRIM28* and Peptidylprolyl Isomerase Like 2 (*PPIL2*), were highly expressed in all surveyed tumor types at both transcriptomic and proteomic levels (Fig. 3a, b), and these E3 ligases may be utilized for PROTAC in a pan-cancer manner, pending a careful survey of their expression in normal tissues, which is the subject of the next section.

Furthermore, the tumor consists of diverse cells, such as malignant, stromal, and immune cells[38], and targets are expected to be degraded by PROTAC primarily in malignant cells. To uncover the expression pattern of E3 ligases in tumors at the single-cell resolution, we collected five single-cell RNA-seq tumor datasets spanning BRCA, NSCLC, UVM, PAAD, and glioma ("Methods"), which were uniformly curated in Tumor Immune Single-cell Hub (TISCH)[39]. We identified the expression pattern varied across E3 ligases within tumors, and

identified several novel E3 ligases that were primarily expressed in malignant cells, suggesting high specificity against targets in malignant cells (Fig. 3c). For example, Lysine Demethylase 5B (*KDM5B*), a potentially novel E3 ligase is highly expressed in breast cancer at both transcriptomics and proteomics, and 98.8% (2448/2478) of cells expressing *KDM5B* are malignant cells in breast cancer (Fig. 3c). 59.7% (2448/4099) of malignant cells expressed *KDM5B*, which is markedly higher than 10.4% (25/241) in immune cells and 14.3% (5/35) in other cells. We also identified potentially novel E3 ligases that were highly expressed at bulk level in tumor and specifically expressed in malignant cells at single-cell levels, such as PRAME Nuclear Receptor Transcriptional Regulator (*PRAME*) in lung cancer and Rho Related BTB Domain Containing 3 (*RHOBTB3*) in Uveal Melanoma (Supplementary Fig. 5a). Recruiting such potentially novel E3 ligases may specifically trigger degradation of malignant cells sparing non-tumor ones.

## The expression landscape of E3 ligases in normal tissues

PROTAC hijacking of highly expressed E3 ligases in normal cells could induce a risk of on-target off-tumor toxicity in undesired normal tissues[1,12,13]. The ideal E3 ligases of PROTAC for cancer therapy will be those highly expressed in tumors but sparingly expressed in normal tissues[4]. Therefore, we examined the transcriptomic expression of E3 ligases in GTEx[40], a pan-tissue transcriptomics atlas of normal samples at the bulk level. We identified a total of 623 (58.0%) E3 ligases that were lowly expressed [log$^{(TPM+1)}$ ≤ 4] in the majority of tissues (≥70% tissues), including 4 existing co-opted E3 ligases, *XIAP*, *VHL*, *DCAF16*, and *AHR* (Fig. 4a, Supplementary Fig. 6a). Notably, *XIAP* did not exhibit high expression in any normal tissues (Fig. 4a, Supplementary Fig. 6a), suggesting a low possibility of off-tumor toxicity. Out of 623 E3 ligases, 619 were not co-opted in PROTAC yet, suggesting the great potential to expand the current E3 ligase pool for degrading targets in tumors at low risk. For example, potentially novel E3 ligase, such as Zinc and Ring Finger 3 (*ZNRF3*) and HECT, C2 and WW Domain Containing E3 Ubiquitin Protein Ligase 1 (*HECW1*), were lowly expressed in all normal tissues (Fig. 4a). Another example is F-Box Protein 6 (*FBXO6*) that were highly expressed in only 2 normal tissues in GTEx (Fig. 4a) and all tumor types in HPA pathology, while the potentially novel E3 ligase also interacts with *EGFR* and Fibroblast Growth Factor Receptor 3 (*FGFR3*) that were well-documented oncogenes[41]. In some situations, though, low expression of E3 ligases is required in a specific tissue or cell type. For example, targeting *BCL-XL*, a platelet-dependent protein, requires low degradation in platelet to avoid toxicity, and a PROTAC was successfully developed by exploiting the low expression of VHL in platelets (NCT04886622). We therefore summarized the E3 ligases that are expressed at relatively low levels in tissues to fulfill such needs, and each tissue has at least 497 E3 ligases that are lowly expressed, indicating a wide choice of selecting an E3 ligase for a single tissue (Fig. 4a).

Furthermore, single-cell RNA sequencing of normal tissues offers unprecedented high-resolution expression profiling, especially for those rare cell populations, to characterize the safety and side-effect of therapy[42,43]. Thus, we further analyzed E3 ligases in Tabula, a pan-tissue single-cell RNA sequencing atlas from normal samples. E3 ligases highly expressed (see threshold in "Methods") in tissues at the single-cell level could be dissected into specific subpopulations. For example, TNF Receptor Associated Factor 3 (*TRAF3*) was highly expressed in colon and skin in Tabula and can be dissected into T cell, B cell, and neutrophil cell of colon tissues and T cells and macrophage of skin (Supplementary Fig. 7a, b). Another example is *CBL* that were highly expressed in liver and can be dissected into endothelial, dendritic cell, and intrahepatic cholangiocyte (Supplementary Fig. 7c). We identified that 966 (89.9%) E3 ligases were expressed low in most normal tissues (≥70% of tissues) at single-cell level. Consistent with the low expression in normal tissues at the bulk level, co-opted E3 ligases, *VHL*, *XIAP*, and *AHR*, showed low expression in all normal tissues in Tabula (Fig. 4a, b,

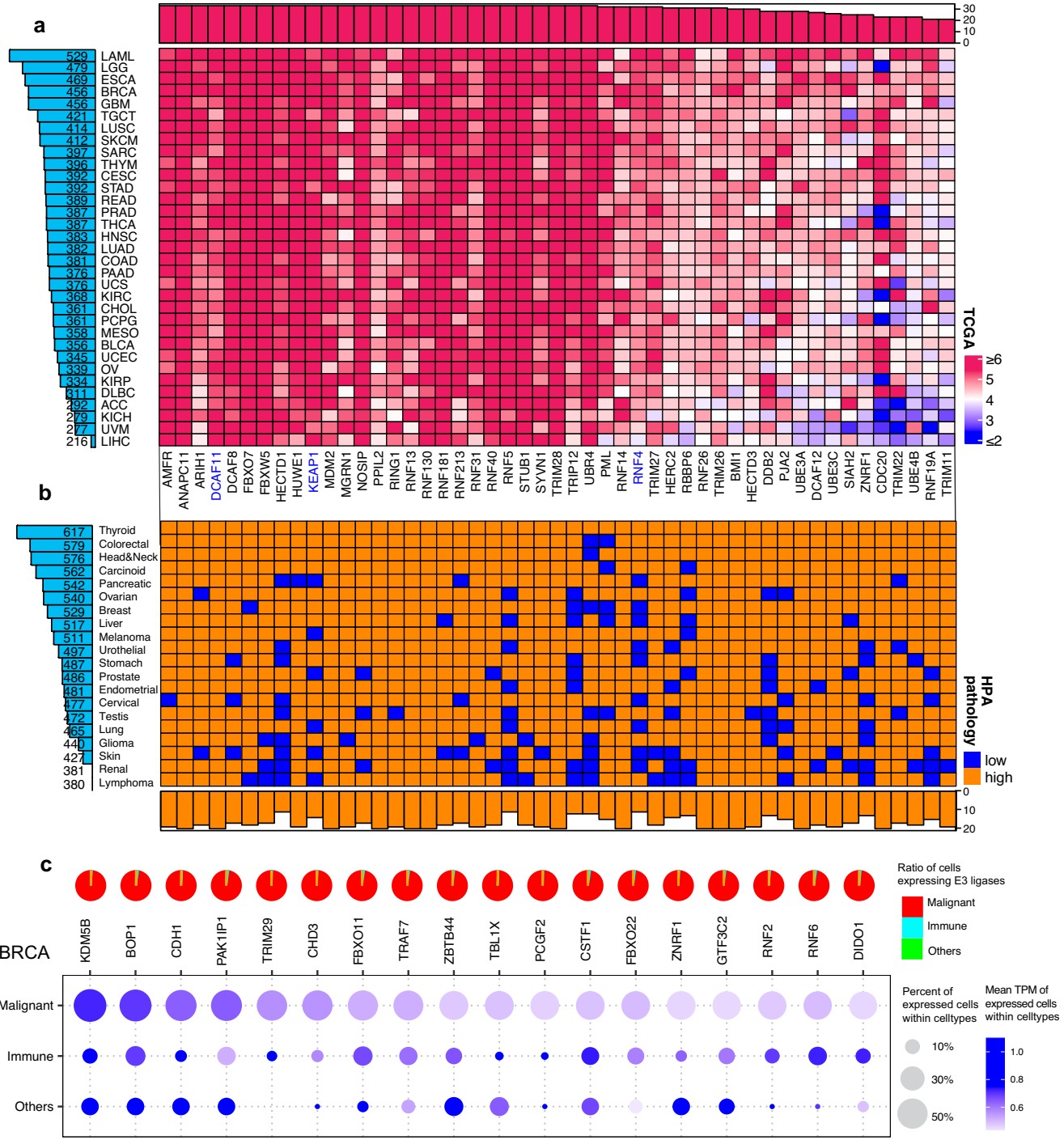

**Fig. 3 | The expression landscape of E3 ligases in tumors. a** Expression of E3 ligases in TCGA. Left side barplot showing the number of E3 ligases highly expressed in the cancer types of TCGA. Top barplot showing the number of highly expressed cancer types of the E3 ligases. **b** Expression of E3 ligases in HPA pathology. Left side barplot showing the number of E3 ligases highly expressed in the cancer types of HPA pathology. Bottom barplot showing the number of highly expressed cancer types of the E3 ligases. **c** Expression of E3 ligases in BRCA cancer type at single-cell level. Pie plot showing the distribution of cells expressing E3 ligases. Dot plot showing the percentage and level of cells expressed E3 ligases within the cell type. E3 ligases in red were proceeded to PROTAC clinical trials. E3 ligases in blue were experimentally explored in PROTAC.

Supplementary Fig. 6b). However, we identified *VHL* has a high-level expression in some tissues, such as liver, blood, lymph node, and thymus in Tabula (Fig. 5b), and subsequent analysis revealed the high-level expression can be dissected into multiple cell types, such as endothelial cell and T cell in liver (Supplementary Fig. 8), suggesting the potential risk of *VHL* in these tissues especially for these cell types, further demonstrating the necessity to develop PROTAC based on novel E3 ligases to circumvent the potential side effect. Beyond these 3

co-opted E3 ligases, 564 novel E3 ligases demonstrated lowly expressed in normal tissues at both bulk and single-cell level, suggesting great potential of novel E3 ligases on reducing toxicity in cancer therapy.

In summary of these large-scale expression screening efforts, through the integration analysis in both tumor and normal tissues, we identified 206 (19.2%) E3 ligases comprised of 3 co-opted and 203 novel E3 ligases that are highly expressed in at least one tumor type in TCGA and HPA pathology and lowly expressed in the majority number

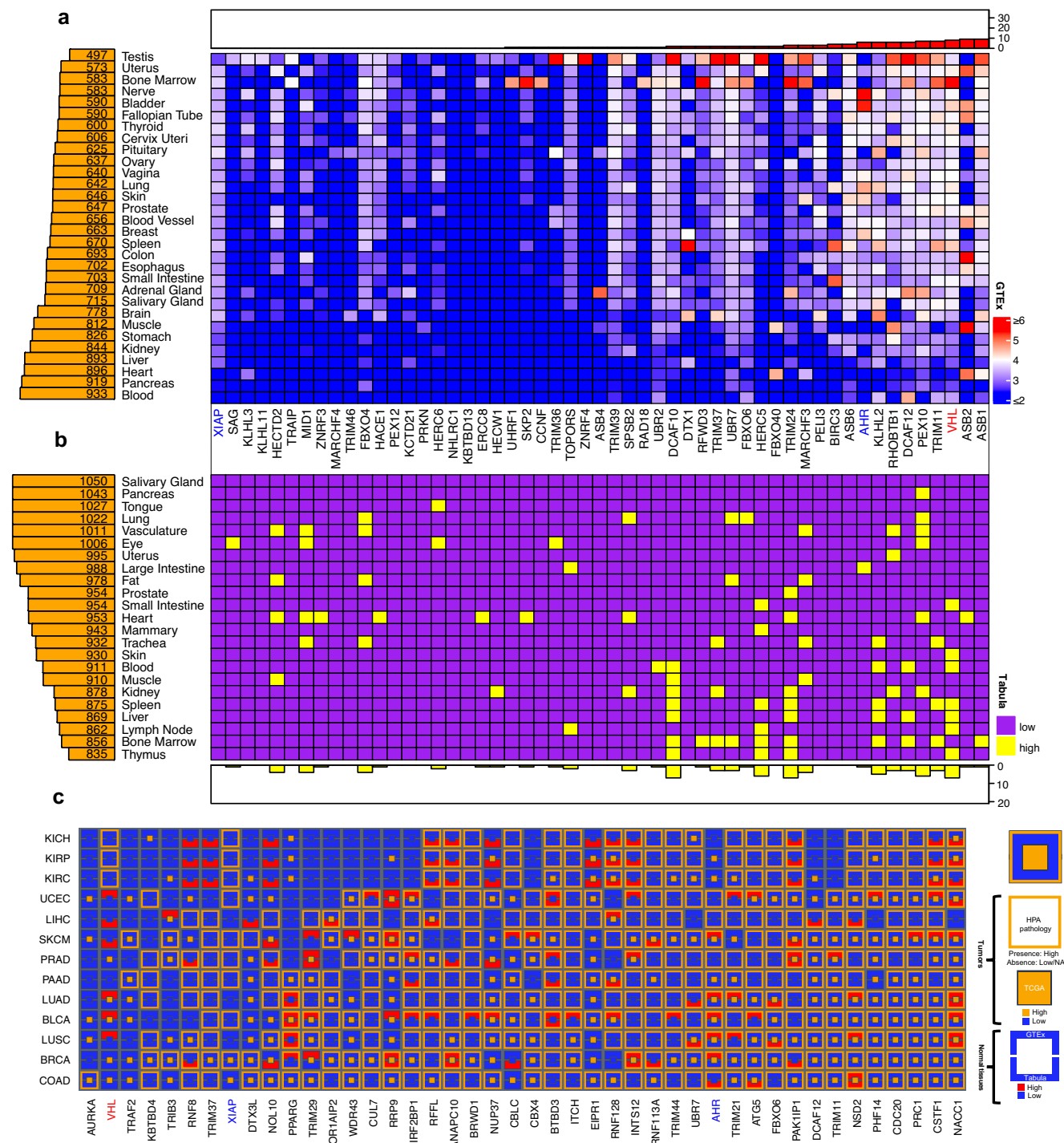

**Fig. 4 | Integrating expression of E3 ligases in normal tissues. a** Expression of E3 ligases in GTEx. Left side barplot showing the number of E3 ligases lowly expressed in the tissue. Top barplot showing the number of high expression of tissues of the E3 ligases. **b** Expression of E3 ligases in Tabula. Left side barplot showing the number of E3 ligases lowly expressed in the tissues. Bottom barplot showing the

number of highly expressed tissues of the E3 ligases. **c** Dissection of expression of E3 ligases featured in four datasets by cancer type. E3 ligases in red were proceeded to PROTAC clinical trials. E3 ligases in blue were experimentally explored in PRO-TAC experiment.

of normal tissues in GTEx and Tabula (Supplementary Fig. 9). The three co-opted E3 ligases, *VHL*, *XIAP*, and *AHR*, have been explored against cancer targets in PROTAC[12,22,44–46], and their expression patterns in tumor and normal tissues suggest extensive future usage, such as *XIAP* in breast cancer (Fig. 4c, Supplementary Fig. 10). Beyond these 3 co-opted E3 ligases, the 203 potentially novel E3 ligases uncovers a large E3 ligase pool to be exploited for anti-cancer PROTAC development. For example, PHD Finger Protein 14 (*PHF14*), interacting with *TP53*[41],

was highly expressed in over 70 percent of cancer types at transcriptomic and proteomic levels in tumors while lowly expressed in most normal tissues. Compared to *VHL*, *XIAP*, and *AHR*, which were highly expressed in less than half of cancer types in TCGA, *PHF14* may serve as a good candidate for pan-cancer usage over co-opted E3 ligases in terms of expression pattern. Another example is Cbl Proto-Oncogene C (*CBLC*), which has evidence of interacting with the major oncogenes, *EGFR* and *ERBB2*[41], and was highly expressed in 12 cancer

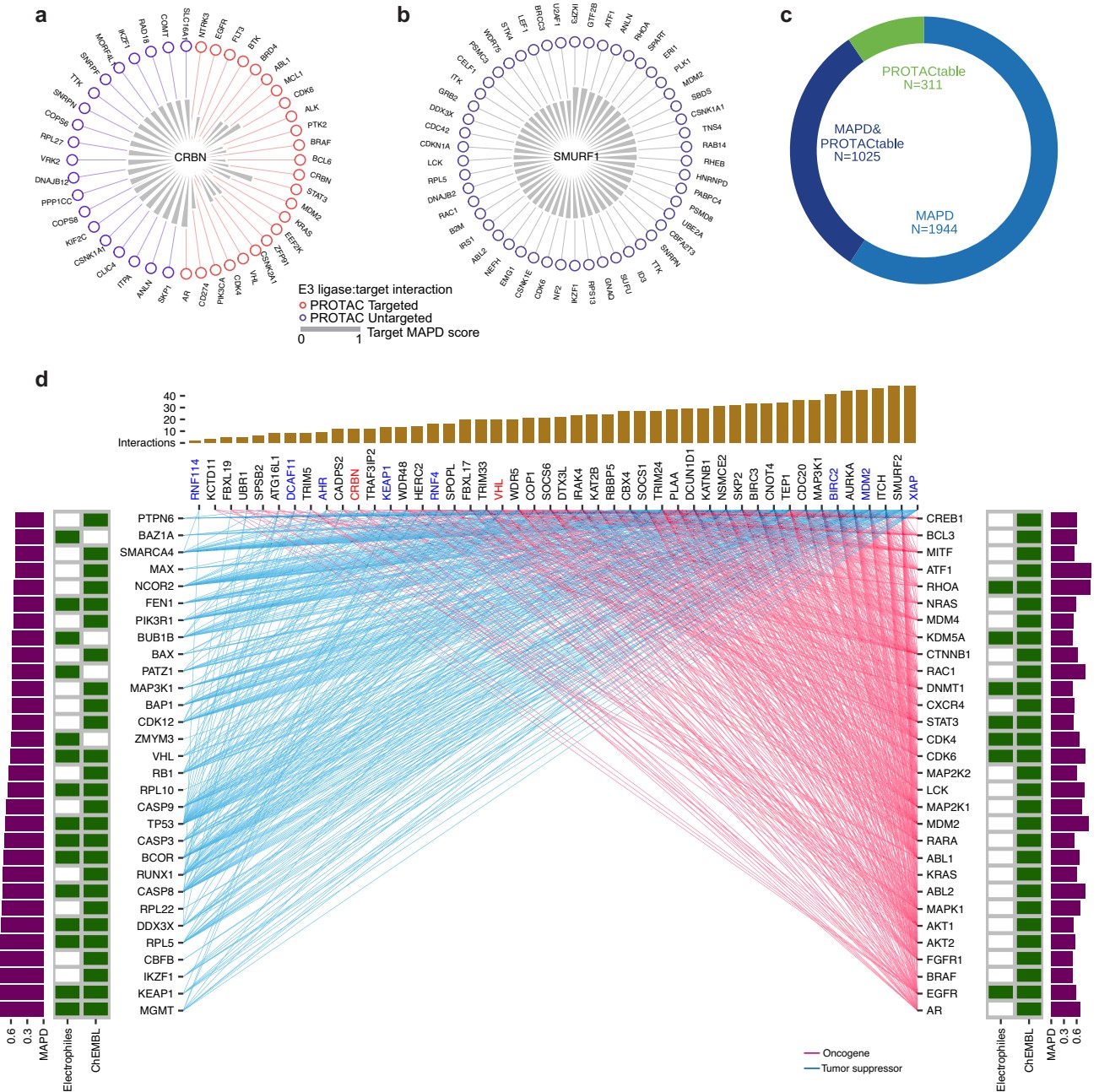

**Fig. 5 | Characterization of PPIs between E3 ligases and targets. a, b** PPIs between E3 ligases and targets. **c** PROTAC tractable targets highlighted in the PROTACtable genome and MAPD in human genome. **d** Interactions between E3 ligases and PROTAC tractable cancer targets. Top barplot showing the number of targets interacting with E3 ligases. Left and right side barplot showing MAPD score of targets. E3 ligases in red were proceeded to PROTAC clinical trials. E3 ligases in blue were experimentally explored in PROTAC experiment.

types at transcriptomic and proteomic levels, and lowly expressed in most normal tissues.

Taken together, we comprehensively delineated the transcriptomic expression at both bulk and single-cell level, and proteomic expression of E3 ligases, demonstrating the necessity to harness the large-scale expression data to further advance PROTAC for higher efficacy and circumventing side effects.

## Comprehensive mapping of protein–protein interactions (PPI) of E3 ligases and POI

The ubiquitination levels and further degradation of targets vary between recruited E3 ligases, and the variations may result from the magnitude of the PPIs between E3 ligases and their targets[16]. Therefore,

we investigated known PPIs to connect E3 ligases and targets. To construct a comprehensive map of PPIs between ligases and potential POI, we collected 1,159,404 unique PPIs between E3 ligases and targets from four types of sources: (1) PPIs involved in literature-based and predicted E3-substrate interactions from UbiBrowser2.0[7]; (2) literature-based of pertaining PPI evidence from BioGrid[47], IntAct[48], and Reactome[49]; (3) artificial intelligence scored PPI evidence from STRING[50]; and (4) assayed PPI evidence from HuRI[51]. We identified a total of 10,930 unique targets having evidence of PPIs with at least one existing co-opted E3 ligase. Effective E3 ligase:target protein interactions of PROTAC in (pre-)clinical trials, such as *CRBN-STAT3* (NCT05225584), *CRBN-AR* (NCT03888612; NCT04428788), and *CRBN-EGFR*[1], can be found in our collected PPIs (Fig. 5a). Expanding co-opted

E3 ligases to all E3 ligases, the number of targets was increased by 76.1% from 10,930 to 19,248, suggesting new E3 ligases may offer great opportunities for degrading novel targets in the proteome through PPI. For example, U2 Small Nuclear RNA Auxiliary Factor 1 (*U2AF1*), a spliceosome gene related to cancers and myelodysplastic syndrome[52,53], has no evidence of PPI with any co-opted E3 ligase but interacts with *SMURF1*, which may serve as a novel E3 ligase in PROTAC (Fig. 5b). Potential new E3 ligase can not only expand potential coverage to novel targets but also provides the opportunity to more specifically against targets. For example, we found a potential novel E3 ligase, *c10orf90*, which interacts with only 4 proteins, including *TP53*, a major cancer driver[54]. Notably, the current co-opted E3 ligases interacting with *TP53*, such as *VHL* and *DCAF11*, interact with at least 200 other targets, which may induce reactions with unexpected targets, and thus *c10orf90* may be a superior candidate to serve as a *TP53*-specific E3 ligase.

PROTACtability (i.e., the likelihood of a protein being degraded by PROTAC) varied across target proteins, and a summary of E3 ligases related PPIs should be prioritized on these targets with high PROTACtability. To identify these PROTACtable targets, we combined the results of two studies, PROTACtable genome[6] and Model-based Analysis of Protein Degradability (MAPD)[55], which performed the assessment of PROTAC tractability of targets. These two studies highlighted 3280 PROTAC tractable targets with an overlap of 1025 targets (Fig. 5c). The median number of co-opted E3 ligases interacting with a PROTAC tractable target is 1, which means most targets can only be identified interacting with 1 co-opted E3 ligase. The median increased to 43 when considering all E3 ligases, indicating utilizing potentially novel E3 ligases may greatly increase the flexibility in selecting E3 ligase targeting a protein (Supplementary Fig. 11). Novel E3 ligases may enable PROTAC against more targets and higher flexibility in selecting E3 ligases. We also identified a number of PPIs involving PROTAC tractable cancer targets that have not yet been explored in PROTAC. For example, Neuroblastoma RAS (*NRAS*), associated with unfavorable prognostic in metastatic colorectal cancer[56], have evidence of PPIs with co-opted E3 ligases *BIRC2* and *XIAP*, and potentially novel E3 ligases *CDC20* and *SKP2* (Fig. 5d). Another PROTAC tractable target is Rac Family Small GTPase 1 (*RAC1*), inducing chemoresistance of breast cancer, interacts with co-opted E3 ligases *MDM2* and *XIAP*, and potentially novel E3 ligases, *ITCH* and *SMURF2* (Fig. 5d). The corresponding E3 ligases may be taken into consideration when developing PROTACs against these cancer targets.

### Structure availability, functional essentiality, cellular location, and PPI interface of E3 ligases

Available structures of E3 ligases could enable the development of small molecule ligands and further recruitment into PROTAC[1] and hence another dimension that is important to consider in their prioritization. To obtain the available protein structures of E3 ligases, we queried the worldwide archive of structure database, Protein Data Bank (PDB)[57]. We identified 414 (38.5%) E3 ligases with available experimentally determined structures, consisting of 9 co-opted and 405 potentially novel E3 ligases (Fig. 6a). These available protein structures offer a great opportunity of recruiting potentially new E3 ligases, such as WD Repeat Domain 5 (*WDR5*) and Embryonic Ectoderm Development (*EED*). Furthermore, proteins exhibit structural plasticity and multiple conformations as dynamic entities[58], and more available structures of E3 ligases could reveal more structural characteristics. Beyond the co-opted E3 ligases, such as *MDM2*, *XIAP*, and *VHL*, which have over 20 available structures, many potentially novel E3 ligases have ample structural information, e.g., Pleckstrin Homology Domain Interacting Protein (*PHIP*) and E1A Binding Protein P300 (*EP300*) (Fig. 6b). Even though some E3 ligases haven't been experimentally resolved, AlphaFold, a high-performance structure prediction framework, has successfully proven to generate high accuracy structures[59],

which makes structure-driven ligand searching and design possible for all potentially novel E3 ligases. For example, Membrane Associated Ring-CH-Type Finger 1 (*MARCHF1*) and Pellino E3 Ubiquitin Protein Ligase 1 (*PELI1*), two potentially novel E3 ligases, don't have available ligands and were not experimentally resolved, but AlphaFoldDB[60] provides predicted structures of the two E3 ligases that may inspire ligand development. We collected the predicted structures of these 661 unresolved E3 ligases from AlphaFoldDB[60] as an alternative source (Fig. 6a).

Due to the central role of E3 ligases in PROTAC, mutations of E3 ligases in tumors could disrupt the degradation function of PROTAC and lead to resistance to the degrader. One potential solution is to employ tumor essential E3 ligases whose genomic alteration results in substantial effects on cellular viability, and the efficacy of PROTAC is less likely to be affected by genomic mutation[1]. We explored the genetic perturbation of E3 ligases via CRISPR and RNAi in Dependency Map (DepMap)[61] that evaluated gene essentiality in cancer cell lines. A recent study revealed CRISPR advantaging at (co-)dependency discovery and high accuracy and RNAi outperforming at identifying associations, and both provide valuable information about cancer dependency[62]. We identified 146 (13.6%) tumor-essential E3 ligases (mean of probabilities of essentiality across cell lines >0.5[63]) in CRISPR screening, including two co-opted E3 ligases, *VHL* and *RNF4* (Fig. 6c, Supplementary Fig. 12). In addition, 144 potentially novel E3 ligases are tumor-essential, suggesting great potential for developing PROTACs using tumor-essential E3 ligases with less vulnerability to genomic loss or deletion. For example, WD Repeat Domain 70 (*WDR70*) and *NEDD8* ubiquitin-like modifier (*NEDD8*) are both essential genes in multiple cancer lines (Fig. 6e). We identified 47 (4.4%) tumor-essential E3 ligases (mean of probabilities of essentiality across cell lines >0.5) in RNAi screening (Fig. 6d). All 47 E3 ligases are potentially novel E3 ligases, and 46 of 47 are also tumor-essential in CRISPR screening. For example, Ring-Box 1 (*RBX1*) which is tumor-essential in both CRISPR and RNAi screening, indicated pan-essentiality in tumor cell lines, and genomic alteration of *RBX1* may possibly reduce cell viability (Fig. 6e). Considering a novel, non-co-opted E3 ligase, *RBX1* showed high essentiality in both screens, suggesting that it may be robustly resistant to emerging inactivating mutations. These potentially novel E3 ligases could be candidate E3 ligases for developing mutation-resistant PROTACs.

Cellular localization is another critical factor in choosing E3 ligases for more precise degradation[1,23]. To identify the cellular location of E3 ligases, we integrated high-quality evidence from COMPARTMENTS[64], Gene Ontology (GO)[65], and UniProt[66]. We identified 983 (87.2%) E3 ligases that were annotated with cytoplasm and/or nucleus, which are considered preferred locations for PROTAC[6] (Fig. 6f). Nine co-opted E3 ligases were identified in both cytoplasm and nucleus, and 3 co-opted E3 ligases were identified solely in nucleus. Our analysis indicated DCAF16 was solely localized in the nucleus, which well aligned with a recently designed PROTAC that employed *DCAF16* for exclusive engagement of nuclear proteins[23]. Beyond co-opted E3 ligases, 572 (53.2%) potentially novel E3 ligases were annotated in nucleus and cytoplasm, such as *BRCA1* and *SMURF1*; 245 (22.8%) potentially novel E3 ligases were identified solely in cytoplasm, such as *KLHL12* and *MIB2*; 166 (15.4%) potentially novel E3 ligases were considered nuclear E3 ligases, such as *DDB2* and *UHRF1* (Fig. 6g). These potentially novel E3 ligases provide a versatile choice for achieving a subcellular localization restricted or required degradation.

Furthermore, E3 ligase:target interactions tend to be disrupted by mutations in the interface between E3 ligase and protein at a higher rate than mutations in non-interface region[67,68]. Due to the difficulty in identifying residues in the interfaces, we collected PPI interfaces from multiple types of sources, including ECLAIR, Interactome3D, and PDB[57,68,69] (Fig. 6h). Collected PPI interfaces span 928 (86.3%) E3 ligases (Fig. 6h), including 11 co-opted E3 ligases (Supplementary Fig. 13).

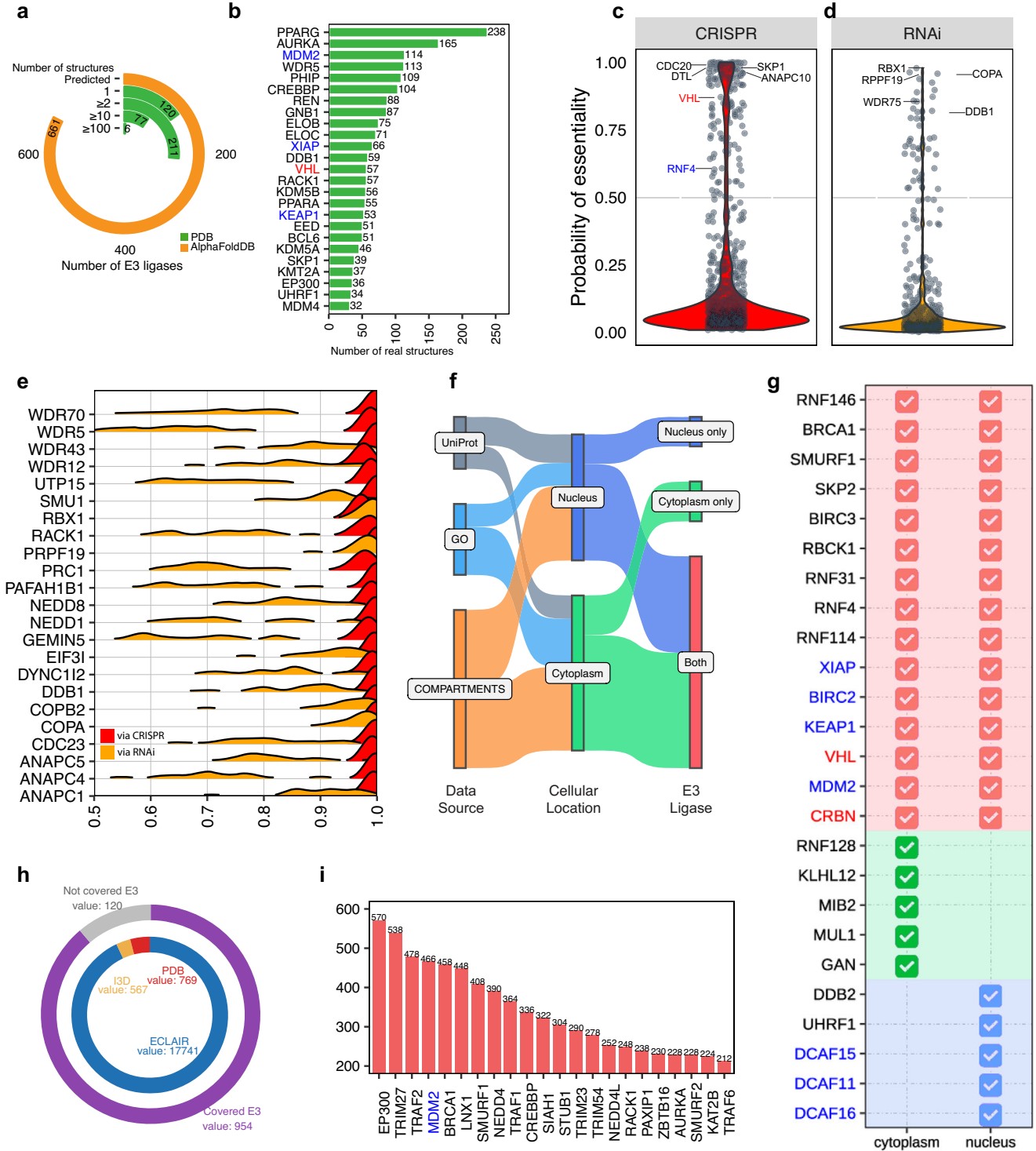

**Fig. 6 | Structure availability, functional essentiality, and cellular location of E3 ligases. a** Distribution of the number of available structures of E3 ligases. **b** The number of protein structures of E3 ligases. **c**, **d** Distribution of mean of the probability of essentiality of E3 ligases by CRISPR and RNAi. **e** Ridgeplot showing the distribution of tumor-essential E3 ligases in both datasets. **f** Sankey plot showing cellular localization of E3 ligases from different data sources. **g** The cellular localizations of E3 ligases. **h** Outer ring showing E3 ligases with available PPI interface and inner ring showing the distribution of origins of PPI interface. **i** E3 ligases with top number of PPI interfaces. E3 ligases in red were proceeded to PROTAC clinical trials. E3 ligases in blue were experimentally explored in PROTAC experiment.

*MDM2*, an experimentally explored E3 ligase, has 466 interfaces, and E1A Binding Protein P300 (*EP300*), a potentially novel E3 ligase, has over 100 more PPI interfaces than *MDM2* (Fig. 6i). This PPI interface information may guide one in determining whether observed mutations in E3 ligases or targets are located in the interface region and thus may further undermine E3 ligase:target interactions.

**A user-friendly web portal of the PROTACtable genome universe of E3 ligases**

We delineated multi-facets of E3 ligases related to PROTAC development and identifying potentially novel E3 ligases for PROTAC development is obviously a complicated decision involving many factors and deep expert knowledge (Fig. 7a). Defining key characteristics in

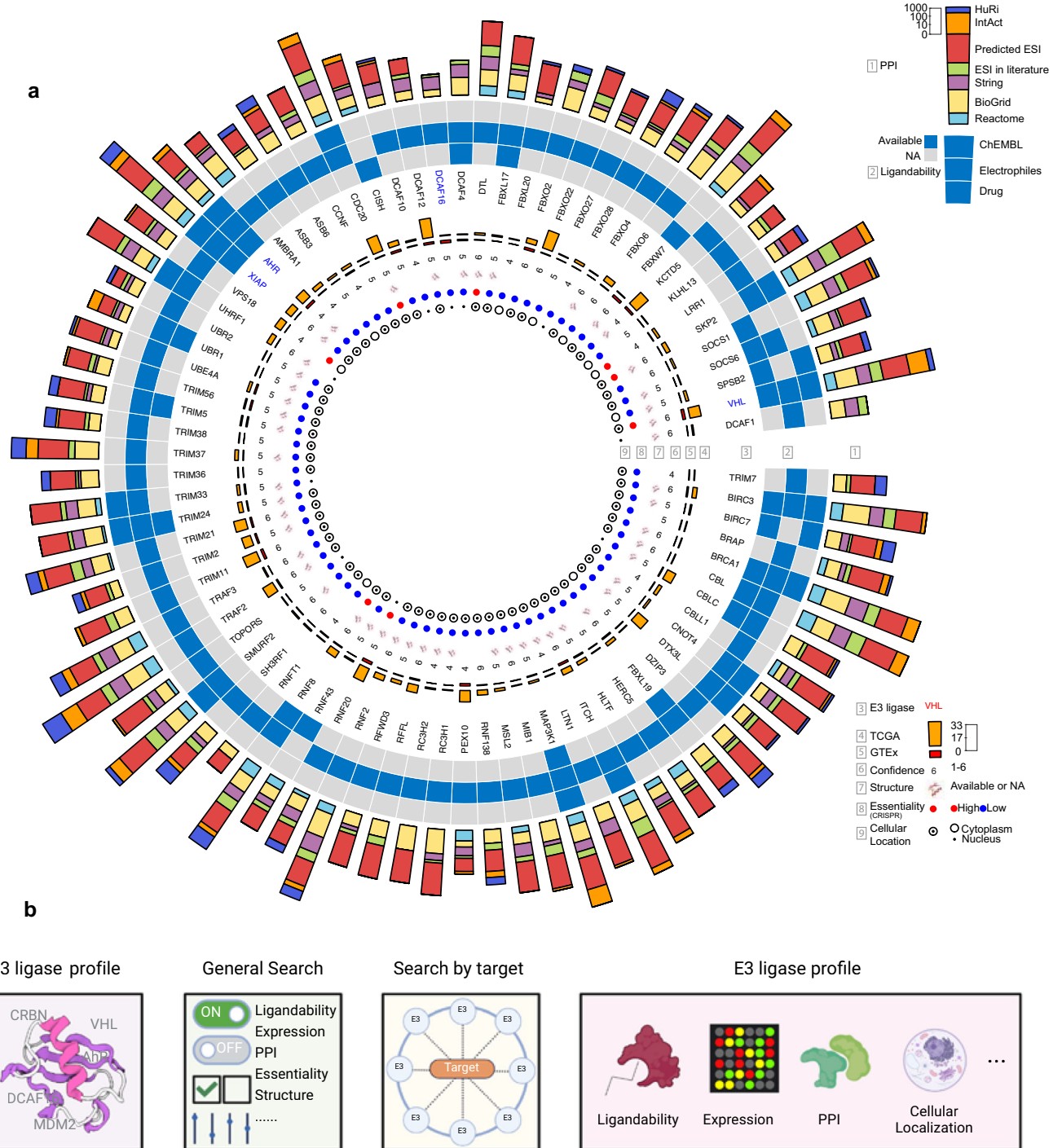

**Fig. 7 | Summary information of E3 ligases and user-friendly web portal. a** From outer to inner ring, PPI, ligandability, E3 ligases, the number of highly expressed cancer types in TCGA, the number of high expression in normal tissues in GTEx, confidence score, structure availability, the probability of essentiality, and cellular localization. E3 ligases in red were proceeded to PROTAC clinical trials. E3 ligases in blue were experimentally explored in PROTAC experiment. **b** Functions and modules of our web portal. Created with BioRender.com.

searching for novel E3 ligase remarkably depends on the specific research questions and challenges. To help domain experts facilitate the search for the best matching E3 ligases given the existing multi-omics large-scale knowledge, we have built a user-friendly data portal (https://hanlaboratory.com/E3Atlas/, Fig. 7b, Supplementary Fig. 14).

Our web portal has 3 modules: (1) E3 ligase profile, (2) General E3 ligase search, and (3) Search by target. The E3 ligase profile module enables users to directly search for an E3 ligase (Supplementary Fig. 14a) to get a comprehensive E3 ligase profile (Supplementary

Fig. 14b). The General E3 ligase search module consists of multiple switches with checkboxes and sliders nested inside to allow users to customize filtering conditions flexibly and precisely (Supplementary Fig. 14c). Switch, checkbox, and slider functions as toggling a group of characteristics, selecting the data sources, and adjusting thresholds, respectively (Supplementary Fig. 14c). For example, turning on the 'Ligandability' switch will retain any E3 ligases having available ligands. Turning on 'Expression in Tumors' switch, checking 'TCGA' checkbox, and moving slider to 70% will retain E3 ligases that are highly expressed

in over 70% of cancer types. The Search by target module enables users to search for E3 ligases that interact with a specific target and specify a cancer type that E3 ligases are highly expressed, and the module also includes options provided in the general E3 ligase search module (Supplementary Fig. 14d). E3 ligases that meet user-defined criteria will be returned as a table (Supplementary Fig. 14e). Recognizing that the PPI of E3 ligases might provide insights into the gene's inherent functions, our web portal incorporates both Gene Ontology and KEGG enrichment analyses on proteins interacting with each E3 ligase.

The applicability and use of the web tool provided are illustrated here in three cases that integrated different sets of factors to identify potentially novel E3 ligases that possess several favorable key characteristics. The first case is searching for E3 ligases that are moderately confident, ligandable, highly expressed in tumors, lowly expressed in the majority of normal tissues, and have evidence of interacting with a number of POIs. Based on these considerations, we designed the following filtering criteria (confidence score >3; available ligands; highly expressed in at least 1 TCGA cancer type and in less than 30% GTEx normal tissues; the number of PPIs >100) and performed the search. We identified 79 E3 ligases that met our defined criteria, including 3 co-opted E3 ligases, *VHL*, *AHR*, and *XIAP*. For example, *VHL* was scored at 6 for its substrate recognition function and has available ligands from ChEMBL, drug, and electrophile (Supplementary Fig. 15). *VHL* also showed a high expression level in most (16 of 33) tumor types and a low expression level in the majority of tissues, and it could interact with 1638 proteins, including *EGFR*, *KRAS*, and *BCL2L1*. Beyond the co-opted E3 ligases, 76 potentially novel E3 ligases also exhibit similar characteristics and may be incorporated into PROTAC development. For example, *SKP2* with a confidence score of 6 for recognizing substrate and mediating ubiquitination, has ligands in ChEMBL and electrophile. *SKP2* was highly expressed in 6 cancer types in TCGA and 1 normal tissue in GTEx, and in collected PPIs it could interact with 2257 proteins (Supplementary Fig. 15). Notably, the 3 derived co-opted E3 ligases were identified in both cytoplasm and nucleus, and we identified 21 potentially novel E3 ligases solely identifiable in either cytoplasm or nucleus, such as *SPSB2* in cytoplasm and *UHRF1* in nucleus, suggesting higher specificity might be achieved by recruiting these potentially novel E3 ligases.

Our analysis also aims to find new E3 ligases targeting key cancer drivers to provide new treatment options. For example, KRAS Proto-Oncogene, GTPase (*KRAS*), the most frequently mutated oncogene[70], has been targeted by PROTACs[71,72], and identifying additional E3 ligases against this critical target remains an important research direction[1]. Considering that *KRAS* frequently drives the pancreas, colon, lung, and other cancers[70], E3 ligases were expected to be highly expressed in these cancer types and also interact with *KRAS*. We identified several potentially novel E3 ligases, such as F-Box Protein 22 (*FBXO22*) and Protein Regulator Of Cytokinesis 1 (*PRC1*), that were highly expressed in multiple cancer types, but lowly expressed in most normal tissues, may serve as good candidates for high expression in above-mentioned cancer types (Supplementary Fig. 16). We also identified a potentially novel E3 ligase that targets *KRAS*, the Potassium Channel Tetramerization Domain Containing 11 (*KCTD11*), which has the lowest number of PPIs among KRAS targeting E3 ligases, suggesting a potential E3 ligase against *KRAS* with higher specificity (Supplementary Fig. 16).

Another crucial oncogene in multiple cancers, such as lung and colon cancers, is Epidermal Growth Factor Receptor (*EGFR*)[73,74]. Our comprehensive collected PPIs revealed that the widely-used E3 ligases, *VHL* and *CRBN*, indeed interact with *EGFR*, in alignment with currently designed EGFR PROTACs[75–77]. However, in total, we identified 28 E3 ligases that interact with *EGFR*, are highly expressed in colon and lung cancers and lowly expressed in the majority of normal tissues. Beyond 3 the known co-opted E3 ligases, we identified 25 potentially novel E3 ligases that also possess these valuable characteristics and target *EGFR*

(Supplementary Fig. 17). For example, F-Box Protein 22 (*FBXO22*), was expressed highly in all cancer types in TCGA and lowly in almost all normal tissues in GTEx, and *CDC20*, showing the highest tumor essentiality in colorectal and lung cancer among all these 28 candidates (Supplementary Fig. 17). These potentially novel E3 ligases may offer alternative choices to the co-opted E3 ligases.

## Discussion

PROTACs provide an alternative way to target proteins that are otherwise considered 'undruggable' by small-molecule inhibitors[4], and recruiting E3 ligase is the necessary step to initiate targeted protein degradation[2]. Despite of the critical role of E3 ligases, only ~1% of E3 ligases have been explored for PROTAC-based protein degradation. Therefore, both academia and industry have repeatedly called for identifying new E3 ligases for precision TPD[1]. To expand the E3 ligase repertoire, we systematically characterized E3 ligases in terms of their fundamental features related to PROTAC development. We characterized the confidence of functioning as an E3 ligase, ligandability, expression pattern, PPI, structure availability, essentiality, cellular location, and PPI interface of E3 ligases. To comprehensively depict E3 ligases, we leveraged a total 30 of large-scale datasets, including three E3 ligase lists, five ligand sources, five expression landscapes, seven PPI databases, two structural data sources, two essentiality screens, three cellular localization resources, and three types of PPI interfaces. Considering the complexity of exploring potentially novel E3 ligases, we integrated these factors and launched a user-friendly web portal to serve the community.

We identified hundreds of potentially novel E3 ligases showing equivalent or even superior characteristics compared to extant co-opted E3 ligases, at least in terms of each feature by comprehensively characterizing these features. Excluding co-opted E3 ligases, we identified 362 potentially novel high scoring E3 ligases, 672 having available ligands, 765 that are highly expressed in cancer, 623 that are lowly expressed in most normal tissues, 923 having PPIs with cancer targets, 405 having experimentally resolved structures, 92 tumor-essential in cancer cell lines, 684 localized in cytoplasm and/or nucleus, and 928 having available PPI interfaces. Combining confidence score, ligandability, expression pattern, and PPI, we identified 76 E3 ligases, that are top PROTAC candidates. Driven by the goal of targeting key cancer drivers, we identified 16 and 28 E3 ligases targeting *KRAS* and *EGFR*, respectively. Beyond facilitating the development of PROTACs targeting cancer, aspects such as confidence scores, ligandability, PPI, structural information, cellular localization, and PPI interfaces of E3 ligases also hold broad applicability in the deployment of PROTACs for a range of other diseases. These factors, therefore, present universal value in the context of PROTAC-oriented therapeutic strategies.

As always, our study has a few limitations. First, we should acknowledge that the lessons learned about E3 ligases in PROTAC are predominantly acquired from previous studies focused on a limited number of well-studied E3 ligases, such as *VHL* and/or *CRBN*. Indeed, as reported in PROTACpedia (https://protacpedia.weizmann.ac.il), which collects active PROTAC molecules, 94.3% (766/812) PROTACs so far recruit either *VHL* or *CRBN*. Second, higher resolution proteomics could provide more accurate insights to evaluate the E3 ligase activity at high resolution, but we included only the bulk level proteomics data, largely due to the currently lacking large-scale, single-cell proteomic data. Thirdly, it is well known that post-translational modifications (PTMs) also play critical roles in regulating the activity and abundance of E3 ligases[78,79]. The abundance of E3 ligases may also be regulated through self-ubiquitination, for example, which has been observed when studying Ubiquitin Protein Ligase E3A (*UBE3A*) and Siah E3 Ubiquitin Protein Ligase 1 (*SIAH1*)[79]. However, the regulatory effects of PTMs on E3s in PROTAC are largely unknown, and large-scale high-resolution PTM data is yet beyond our reach[80]. Bearing these limitations in mind, we expect to continue and update our web portal as

more knowledge and data are gathered, and we imagine that this will need to be done repeatedly every few years in the future.

In the future our approach could be potentially be extended to other UPS-based target degradation techniques, in which E3 ligases play a critical role. One promising domain involves PROTABs, which hijack cell-surface E3 ligases via antibodies to degrade transmembrane proteins. In designing effective PROTABs, the characteristics of E3 ligases, such as the expression pattern, are also crucial factors in choosing the most suitable E3 ligase[18]. For example, *RNF43* was employed for its high expression level in colon adenoma compared with normal tissue in PROTAB development[18], and our analysis also highlighted *RNF43* that was highly expressed in 2 TCGA tumor types, including Colon adenocarcinoma (COAD), and lowly expressed in all GTEx normal tissues, suggesting our analysis could also support the development of other related TPD fields. Further future extensions may be applied to study molecular glues, which induce interaction between E3 ligases and a target protein as a single molecule[81]. In this context, our analysis could provide the characteristics of E3 ligases themselves, such as expression pattern, essentiality, and cellular location, which are important factors in their rationale-based selection.

PROTACs effectively suppress protein activity, but when they target essential proteins, on-target toxicity may arise[82,83]. To minimize this, PROTACs can be designed to interact with E3 ligases that are selectively expressed in specific tissues or cell-types, mitigating undesired effects[83]. As an example, *VHL*'s unique expression was leveraged in *BCL-XL*-targeting PROTACs to reduce toxicity[12]. As other novel treatments like immunotherapy may lead to serious toxicity[37,43,84,85], and with the first PROTAC entering clinical tests in 2019, we expect more toxicity reports. Thus, we developed a platform to evaluate such potential on-target, off-tumor effects and aid in toxicity monitoring.

Our study built upon insights from prior research to identify new E3 ligases that could be integrated into PROTAC development. As future experiments unveil more insights, we could identify additional factors and resources to provide more precise guidance on the search for new E3 ligases. In summary, we expect that the results of this first-of-its-kind multi-dimensional comprehensive analysis, summarized in a user-friendly flexible web portal, will markedly enhance the ability of PROTAC researchers to rapidly identify new E3 ligases with promising TPD activities against specifically desired targets, facilitating the development of these therapies in cancer and beyond.

## Methods

### E3 ligases collection
We carefully selected three E3 ligase lists that focused on different characteristics of E3 ligases, including Ge et al.[9], UbiHub[8], and UbiBrowser2.0[7]. All gene symbols were mapped to HUGO symbol[86] (https://www.genenames.org/) for consistency. We calculated a cumulated confidence score of each E3 ligase by looking up records in three lists. In Ge et al., a score of 2 was assigned to these E3 ligases that were validated in the literature, and the predicted E3 ligases received a score of 1. In the UbiHub, E3 ligases whose function annotation contains the keyword 'E3' received a score of 2 otherwise 1. In the Ubi-Browser2.0, E3 ligases with over 5 E3-Substrate Interactions (ESIs) were assigned a score of 2 otherwise 1. For an E3 ligase not included in the given source, it wouldn't receive a score. E3 ligases explored in PRO-TACs were obtained in PROTAC-DB[5]. BIRC2 and XIAP were retained to represent IAPs because these two are believed to be mainly used by IAP-based PROTACs[46]. In addition to E3 ligases reported in PROTAC-DB, another E3 ligase, KEAP1, was reported to be adopted in PROTAC from a recent study[21].

### Ligandability
Ligandability information was summarized and aggregated from multiple experimental chemical databases. Each record related to E3

ligase in ChEMBL[27] (https://www.ebi.ac.uk/chembl/), the comprehensive manually curated chemical database, was screened. Interactions between small-molecule ligand and target with pChEMBL ≥ 5 (i.e., negative ten-based logarithm of IC50, XC50, EC50, AC50, Ki, Kd or Potency) were considered active and collected[6]. Drugs interacting with E3 ligase gene/gene-product were obtained from DrugBank[25] (https://go.drugbank.com/) and DGIdb[26] (https://www.dgidb.org/). Due to the duplicate records across databases, all chemicals were mapped to ChEMBL for unique id if applicable, and redundant records were discarded. Beside non-covalent ligands, covalent ligand has attracted more interest since electrophilic PROTACs successfully induced neo-substrate degradation[23,87]. A recent study performed a large-scale proteome-wide screening for reactive cysteine via streamlined cysteine activity-based protein profiling[28] (http://wren.hms.harvard.edu/cysteine_viewer/). The resource comprising 285 electrophiles with three human cell lines was adopted to render the covalent ligandability of E3 ligases. E3 ligases associated with drugs in DrugBank and DGIdb were designated as "E3drug." Those linked to small molecules in ChEMBL were marked as "E3chem" if no drug information was available. E3 ligases associated with covalent ligands were labeled as "E3cova" when no drug or small molecule information was present. E3 ligases lacking any ligand information were categorized as "E3dark."

Drug-target interactions were predicted with a deep-learning-based model, HyperAttentionDTI[31]. Protein sequences fed into the model were obtained from UniProt[66]. Drug Simplified Molecular Input Line Entry System (SMILES) structures were obtained from DrugBank[25]. HyperAttentionDTI provided a DrugBank dataset[31], briefly containing balanced 35,022 drug-target interactions and split into training, validation, and testing sets in a 16:4:5 ratio. We performed the evaluation with four configurations: (1) both drugs and proteins in the testing set are present in the training sets; (2) drugs in the testing set are excluded from the training set; (3) proteins in the testing set are excluded from the training set; and (4) both drugs and proteins in the testing set are absent in the training set. These setups range from traditional leave-ligands-out and leave-protein-out methods to a stringent test ensuring that neither drug nor protein was seen during training.

### Gene and protein expression
The Cancer Genome Atlas (TCGA) and Human Protein Atlas (HPA) pathology[36] collected tumor samples and provided transcriptomics and proteomics expression profiles at bulk level, respectively. Expression levels of E3 ligases in normal samples at bulk level were retrieved from GTEx[40] and HPA mRNA[88]. To further precisely delineate expression pattern at single-cell level, Tabula Sapiens[89] was obtained to characterize the expression in normal and tumor samples.

Transcriptomic expression of TCGA and GTEx samples was downloaded from the UCSC Toil recompute[90] (https://xenabrowser.net). In Toil recompute, transcriptomic expression of all samples was analyzed using a single script to achieve consistency, and the unified pipeline consists of adaptor cutting, alignment, and quantification. TCGA and GTEx tissue expression were defined as the median of $\log_2(TPM+1)$ across samples. If the expression level is higher than 4, then the tumor type or tissue was denoted as high expression[37]. Protein expression of tumor samples was downloaded from Human Protein Atlas Pathology section (https://www.proteinatlas.org/humanproteome/pathology), and protein expression data were generated by immunohistochemically staining. We used median expression level analysis to minimize outlier effects.

Genes with available HPA pathology expression were classified as 'High' for whose median-high expression was identified in over 20 percent of samples, otherwise as 'Low'[37].

Single-cell RNA sequencing expression datasets were downloaded from TISCH (http://tisch1.comp-genomics.org/), and TISCH performed all analyses using a uniform streamlined processing to minimize batch effects and annotate cell type consistently[39]. Studies were retained by

the following criteria: (1) sampling from humans; (2) treatment naïve; (3) assayed via 10X Genomics; (4) malignant cells annotated. When multiple studies shared with the same cancer type, the study with the largest number of samples was retained for the subsequent analysis. We finally obtained datasets included BRCA[91], Glioma[92], NSCLC[93], PAAD[94], and UVM[95].

Single-cell transcriptomic expression of normal tissues was retrieved from Tabula Sapiens Portal (https://tabula-sapiens-portal.ds. czbiohub.org)[89]. Tabula Sapiens used a consistent protocol in sample collecting, processing, analyzing, and quality control. All cells comprising 24 tissues in Tabula Sapiens were retrieved, and due to the discrepancy between sequencing platforms, only cells assayed by 10X were retained for subsequent analysis. Due to the lack of overexpression threshold of E3 ligases in PROTAC, we referred the expression of VHL in platelets, which was believed a safe case[12]. In our preliminary analysis, VHL was expressed in 17% of cells at a mean of 3.1 in non-zero platelets in Tabula, and we adopted 75% of this expression pattern, which is the more stringent criterion to identify potential side effects. For a specific gene, tissues with over 12.75 percent of non-zero expressed cells and mean expression over 2.325 were considered high-expression tissue, otherwise as "Low". Matching tumor type and corresponding normal tissues was referred to a recent study[96].

## Protein–protein interactions (PPI)

General PPIs between E3 ligase and target protein were collected from BioGrid[47] (https://thebiogrid.org/), IntAct[48] (https://www.ebi.ac.uk/intact/), and Reactome[49] (https://reactome.org/), HuRI[51] (http://www.interactome-atlas.org/), STRING[50] (https://string-db.org/). To ensure the quality of the incorporated PPIs, we extracted PPIs from literature curation sources such as BioGrid, IntAct, and Reactome, carefully removing any non-human entries. We marked PPIs that were in physical association identified via methods such as yeast two-hybrid studies, affinity purification-mass spectrometry, protein 3D structures, or low-throughput experiments, serving as indicators for users that these PPIs were obtained in a high confidence. Entries from HuRI were retained due to their comprising solely of experimentally verified human binary protein interactions. PPIs from STRING were combined with known and predicted PPIs, and we only retained those human-related and with high confidence (score >700). PPIs specific for E3-substrate interactions (ESIs) were retrieved from from UbiBrowser2.0[7] (http://ubibrowser.bio-it.cn/), retaining only human ESIs documented in literature and high-confidence predicted ESIs with scores exceeding 0.7. NCG[41] (http://ncg.kcl.ac.uk/) was used to annotated the target for its role in cancer. PROTAC tractability of targets were adopted from PRTOACtable genome[6] and MAPD[55]. The interfaces between proteins were obtained from InteractomeInsider that combined cocrystal structures from PDB[57], homology models from Interactome3D[69], and predicted interface by InteractomeInsider core algorithm ÉCLAIR (high confidence only)[68]. Enrichment of gene ontology (GO) and Kyoto Encyclopedia of Genes and Genomes (KEGG) on proteins interacting with E3 ligases were tested using a hypergeometric test in the clusterProfiler4.0 package[97], and p values were adjusted with the Benjamini and Hochberg procedure. Terms with adjusted p values < 0.05 were considered significant.

## Structures of E3 ligases

Available experimental resolved protein structures were obtained in PDB[57] according to UniProt ID mapping (https://www.uniprot.org/id-mapping). Proteins without available structures were referred to AlphaFold Protein Structure Database[98] (https://alphafold.ebi.ac.uk/).

## Essentiality

CRISPR screens of E3 ligases were downloaded from DepMap[61] (https://depmap.org/portal/download/, 22Q2 release). RNAi screens of E3 ligases were obtained from DepMap RNAi project[99]. The mean

probability of dependency of the same cell line lineage was calculated to determine the dependency of E3 ligase for a given cell line lineage. The mean probability over 0.5 were denoted as "essential", otherwise as "not essential"[63].

## Cellular location

High-confidence cellular location was obtained from Gene Ontology (GO)[65] (http://geneontology.org/), COMPARTMENTS[64] (https://compartments.jensenlab.org/), and UniProt (https://www.uniprot.org/). The high confidence evidence in GO and UniProt was defined according to PROTACtable genome[6]. High confidence annotation (score > 3) from COMPARTMENTS was included. Favorable cellular locations were defined in PROTACtable genome[6], including cytoplasm (annotated with "cytoplasm" and/or "cytosol") and nucleus (annotated with "nucleus"). E3 ligase with a cellular location from any of the sources was considered distributed in the location.

## PPI interface

PPI Interfaces were downloaded from InteractomeINSIDER (http://interactomeinsider.yulab.org)[68]. Three types of PPI interfaces, interface derived from PDB[57], Interactome3D[69], and high-confidence ECLAIR model[68] were obtained for subsequent analysis.

## Web portal development

The web portal was developed using Python Flaks framework[100] and deployed in JetStream2 Cloud system[101,102].

## Reporting summary

Further information on research design is available in the Nature Portfolio Reporting Summary linked to this article.

## Data availability

This study made use of publicly available datasets. In order to collect a comprehensive E3 ligase list, data from Ge et al.[9], UbiHub (https://ubihub.thesgc.org/static/UbiHub.html)[8], and UbiBrowser2.0 (http://ubibrowser.bio-it.cn)[7] were sourced. Ligandability of E3 ligases were derived from ChEMBL (https://www.ebi.ac.uk/chembl)[27], DrugBank (https://go.drugbank.com)[25], DGIdb (https://www.dgidb.org)[26], and streamlined cysteine activity-based protein profiling (http://wren.hms.harvard.edu/cysteine_viewer)[28]. Gene expression matrices of the TCGA and GTEx were obtained in UCSC Toil Recompute Compendium (https://xenabrowser.net)[90]. Protein expression levels in tumors were downloaded from HPA pathology atlas (https://www.proteinatlas.org/humanproteome/pathology)[36]. Single-cell transcriptomics data in normal samples were fetched from the Tabula Sapiens portal (https://tabula-sapiens-portal.ds.czbiohub.org)[89], and data in tumors were acquired in the TISCH database (http://tisch1.comp-genomics.org)[39]. Matched public single-cell expression of tumor samples were originally from GSE143423 for BRCA, EMTAB6149(NSCLC for NSCLC, GSE139829) for UVM, CRA001160 for PAAD, GSE138794 for glioma. PPIs and ESIs were collected from BioGrid (https://thebiogrid.org)[47], IntAct (https://www.ebi.ac.uk/intact)[48], Reactome (https://reactome.org)[49], HuRI (http://www.interactome-atlas.org)[51], STRING (https://string-db.org)[50], and UbiBrowser 2.0 (http://ubibrowser.bio-it.cn)[7]. The role of target gene in cancer was annotated with NCG[41] (http://ncg.kcl.ac.uk/). Protein–protein interaction interfaces was downloaded from InteractomeInsider (http://interactomeinsider.yulab.org/)[68]. Structures of E3 ligases were queried and summarized from the PDB (https://www.rcsb.org/)[57] and AlphaFoldDB (https://alphafold.ebi.ac.uk/)[60]. Essentiality of genes in tumors were obtained from the DepMap via CRISPR method (https://depmap.org/portal)[61] and an RNAi-based study[99]. Cellular locations were determined using the Gene Ontology database (http://geneontology.org)[65], COMPARTMENTS (https://compartments.

jensenlab.org)[64], and UniProt (https://www.uniprot.org)[66]. Processed data was deposited into our web portal (https://hanlaboratory.com/E3Atlas). Source data are provided with this paper.

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

## Acknowledgements

Research reported in this publication was supported by the National Institutes of Health (NIH) (R01HG011633 and R01CA262623 to L.H.; R21CA277257 to Y.Z.). The content is solely the responsibility of the authors and does not necessarily represent the official views of the National Institutes of Health. The work was also supported by the Leukemia & Lymphoma Society (to Y.Z.), and the Welch Foundation (BE-1913-20220331 to Y.Z.). E.R.'s research was supported in part by the Intramural Research Program of the NIH, National Cancer Institute and the Center for Cancer Research.

## Author contributions

L.H. conceived and supervised the project. Y.L. designed and performed the research. Y.L., J.Y., T.W., M.L., Y.C., and C.C. performed the data analysis. Y.L., Z.R., Y.Z., E.R., and L.H. wrote the manuscript with input from all other authors.

## Competing interests

Eytan Ruppin is a co-founder of Metabomed Ltd and MedAware, and a (divested) co-founder and non-paid scientific consultant for Pangea Biomed. The remaining authors declare no competing interests.
