## [Peer Review File · Nature Communications]

Re: NCOMMS-23-12374-T

REVIEWER COMMENTS

Reviewer #1 (Remarks to the Author):

Liu et al describe a project in which data on E3 ligases from various sources are integrated, with a view to assisting the selection of new/novel members of that family for example in PROTAC development. The integrated information is made available via a public website. This work would be of interest to the broader protein degradation community and is worthy of publication. However, I don't consider it to contain sufficient "noteworthy" original results to warrant publication in Nature Communications. In some respects it is an incremental advance on previous work such as the PROTACtable Genome analysis of Schneider et al; furthermore the paper does not provide any particularly insightful analyses of the E3 ligase landscape beyond some illustrative examples throughout the manuscript (none of which have any experimental validation). It would seem to me that a more suitable home for such a study would be the companion Nature journal Scientific Data (though this is ultimately a decision for the editor).

Response: PROTACs employ E3 ligases to tag target proteins for subsequent degradation, and the recruitment of E3 ligases is an essential step in the process of target protein degradation¹. The PROTACtable Genome analysis of Schneider et al is indeed a comprehensive and very worthy analysis focusing on the target proteins of PROTACs², but it did not consider the potential of new E3 ligase that could be co-opted into the PROTAC development. Furthermore, the prevalent use of VHL and CRBN has presented a challenge to the current development of PROTACs³, thereby motivating our search for novel potential E3 ligases. Please note, our work is the first comprehensive analysis focusing on the features of E3 ligases in the PROTAC, and in this respect is clearly not an incremental advance compared to previous work, e.g., PROTACtable genome analysis.

To further explicate why this is indeed the case, we note that our study has carried out a comprehensive evaluation of all E3 ligases: to this end we collected 30 datasets covering different aspects of PROTAC, including chemical ligandability, expression patterns, protein-protein interactions (PPI), structure availability, functional essentiality, cellular location, and PPI interface. We then systematized and ranked the E3 ligases that could be prioritized into PROTAC development. We further provide a few examples to illustrate the methodology of identifying potent E3 ligases that could act against key cancer targets. Finally, we developed a user-friendly flexible web portal (<https://hanlaboratory.com/E3Atlas>) that is aimed at assisting the broader protein degradation community to rapidly identify new E3 ligases with promising TPD activities against specifically desired targets, facilitating the development of PROTAC in cancer and beyond.

An important novelty of our work relates to the computational-centralized focus. Our approach follows recent studies that have established the significance of rigorous analysis which is based on large-scale data. For example, the recent TCGA/ICGC Pan-cancer project published a series of high-profile papers in Nature and Cell series journals⁴⁻¹¹. These comprehensive analyses provide the foundation and important insights for further investigations, which may be performed

independent of experimental validation (as demonstrated in studies from our group in the cancer genomics field¹²⁻¹⁶).

I do have serious concerns about the inclusion of output from predictive models alongside data from ‘real’ (often peer-reviewed) experiments. The ‘hyperattentionDTI’ model in particular in my view provides results that are of no real value and indeed may well be misleading. An average of 552 drug-ligase interactions is frankly unbelievable given the typical number of meaningful protein:ligand interactions of marketed drugs for example from chemoproteomics experiments. I would advise the authors to seek out someone with expertise in the practical development, validation and use of in silico models in drug discovery (preferably from an industrial background) who will be able to advise on how best to utilise such methods.

Response: We thank for this valuable comment. We reevaluated our method, and adopted the data presentation format from the original HyperAttentionDTI study, which highlights predicted interactions with the highest probability as potential candidate interactions¹⁷. Addressing your concern, in the revised manuscript, we now present only the top predicted drug-target interactions for each E3 ligase. By default, the top 10 predicted interactions are displayed, and users have the flexibility to expand this list from 1 to 500 at their choice. Furthermore, we also included an option that allows users to choose a threshold to examine predicted interactions meeting a specific cutoff.

The Screenshot of virtual screening of data portal has been modified accordingly:

Identifier	Label	E3 Ligase	Probability (0-1, 1 highest)
DB11327	Yes	TRIM24	1
DB00812	Yes	TRIM24	1
DB04400	Yes	TRIM24	0.9949
DB00991	Yes	TRIM24	0.9939
DB12010	Yes	TRIM24	0.9913

We revised the main text accordingly, which now reads (page 7): “To search for the potentially new drugs for recruiting E3 ligases, we adopted a deep-learning based virtual screening ligand model, HyperAttentionDTI¹⁷, to search for potential interactions between E3 ligases and drugs in DrugBank¹⁸. The model captured complex interactions between atoms and amino acids through an innovative attention mechanism and achieved improved performance over the state-of-the-art baselines, and the new drug-target interactions falls into top virtual screening predicted interactions in the test case study. After evaluating and applying this model (Figure S3), we obtained predicted drug-target interactions, including for potentially novel E3 ligases, such as Makorin Ring Finger Protein 1 (MKRN1), having PPIs with key tumor drivers, TP53 and APC32, and predicted to

interact with DB13955 (Estradiol dienanthate) and DB03017 (Lauric acid). Utilizing these drugs may increase the ligandability of potentially novel E3 ligases that target additional tumor targets. Taken together, our systematic analysis exploited five large-scale data sources and revealed a number of new E3 ligases with tens of available ligands that may be utilized for PROTAC development.”.

The authors seem to be in two minds as to whether they are addressing a cancer audience (which would be entirely reasonable given their own areas of research focus) or a wider, disease-agnostic audience (as implied in the Introduction).

Response: Thanks. Considering the wide use of PROTAC in many areas, such as cancer, immune disorder, and neurodegenerative diseases¹⁹, we aimed at providing insights for broader fields beyond cancer. Accordingly, many of E3 ligases properties carefully evaluated in this manuscript are general characteristics involved in PROTAC development. We clarified this in the revised manuscript (page 18-19), stating that “Beyond facilitating the development of PROTACs targeting cancer, aspects such as confidence scores, ligandability, PPI, structural information, cellular localization, and PPI interfaces of E3 ligases also hold broad applicability in the deployment of PROTACs for a range of other diseases. These factors, therefore, present universal value in the context of PROTAC-oriented therapeutic strategies.”

Some additional caveats concerning the real-world relevance of certain data types may also be in order and worthy of comment or more in-depth analysis. For example, how high in reality is the risk of on-target off-tumour tox with clinical degraders? Especially as, by design, PROTACs are not necessarily very potent inhibitors at the relevant E3 ligase.

Response: We thank for this insightful comment. PROTACs can effectively suppress protein activity, and on-target toxicity has become one of the major concerns when these target proteins are essential in normal tissues^{20,21}. To mitigate the on-target toxicity of PROTAC, one efficient method is to recruit E3 ligases that are selectively expressed in desired cell types or tissues. As a result, the PROTAC is less likely to exert its function in the undesired cells or tissues²¹. For example, when developing BCL-XL-targeting PROTACs, VHL was deliberately selected to relieve the on-target toxicity due to VHL's unique expression pattern²². Novel advanced cancer therapeutics, such as CAR-T therapy, were reported to have serious and potentially life-threatening toxicities when more products move into clinical trials^{23,24}. With the first PROTAC entering clinical trials in 2019 and more on the way, we expect more cases of toxicity will be reported. Thereby, our works has firstly tried to establish a systematic platform in order to circumvent on-target, off-tumor toxicity and recommend close monitoring on potential signs of toxicity. We clarified this in the revised manuscript as follows: “PROTACs effectively suppress protein activity, but when they target essential proteins, on-target toxicity may arise^{20,21}. To minimize this, PROTACs can be designed to interact with E3 ligases that are selectively expressed in specific tissues or cell-types, mitigating undesired effects²¹. As an example, VHL's unique expression was leveraged in BCL-XL-targeting PROTACs to reduce toxicity²². As other novel treatments like immunotherapy may lead to serious toxicity²³⁻²⁶, and with the first PROTAC entering clinical tests in 2019, we expect more toxicity reports. Thus, we developed a platform to evaluate such potential on-target, off-tumor effects and aid in toxicity monitoring.” (page 20).

I found the specific references to PPARG and AURKA rather odd; are these really confirmed E3 ligases? Furthermore, when I tested the website using these two proteins the system appeared to 'hang' and provided no results, which did not inspire much confidence.

Response: Collecting a list of E3 ligase is challenging due to the diverse and integrative nature of the ubiquitin pathway²⁷. We therefore assembled credible E3 ligase lists from three different sources, including UbiHub²⁸, Ge et al.²⁷, and UbiBrowser2.0²⁹, and assigned a confidence score (1-6). This was done to capture a comprehensive list of candidate E3 ligases as is currently best possible.

Regarding the E3 ligase functions of two specific genes, PPARG and AURKA, both genes were collected in Ge et al. 2018 and UbiBrowser as E3 ligases. PPARG was considered as E3 ligases in Ge et al. based on literature³⁰, and in UbiBrowser based on literatures³¹⁻³³. AURKA was considered as E3 ligase in Ge et al. based on literature³⁴, and in UbiBrowser based on literatures³⁵. This evidence suggests that PPARG and AURKA are very likely function as an E3 ligase.

Regarding to the display issues, we apologize for that and have carefully resolved the issue and any other remaining issues, to the best of our knowledge.

That aside, the website did seem to function as advertised for more established E3 ligases and was straightforward to use. I would however like to see an option to download the entire dataset for exploration using a user's own data analysis tools.

Response: Thanks. We have now added a download tab to allow users process the data at their own preference.

Screenshot of download page:

E3 Ligase Landscape Home E3 Profile General Search Search by Target Document Download Contact

Download

Given the intricacies of data structure and the necessity for efficient network transmission, the data is housed in the Feather format. Feather is a highly efficient, fast columnar data format compatible with several programming languages, including Python and R.

E3 ligase general information

Expression

- E3 ligase - TCGA
- E3 ligase - GTEx
- E3 ligase - HPA pathology
- E3 ligase - Tabula
- E3 ligase - TISCH

HyperAttentionDTI

- E3 ligase - HyperAttentionDTI

PPI

- E3 ligase - PPI

Dependency

- E3 ligase - CRISPR based
- E3 ligase - RNAi based

There were also a few relatively minor typos that will need to be fixed before publication.

Response: We have revised our manuscript and corrected typos in the revised manuscript. For example:

The ideal E3 ligases of PROTAC for cancer therapy will be those highly expressed in tumors but sparingly expressed in normal tissues³⁶. (page 9)

The three co-opted E3 ligases, VHL, XIAP, and AHR, have been explored against cancer targets in PROTAC^{22,37-40}, and their expression patterns in tumor and normal tissues suggest extensive future usage, such as XIAP in breast cancer (Figure 4C & Figure S10). (page 11)

Expanding co-opted E3 ligases to all E3 ligases, the number of targets was increased by 76.1% from 10,930 to 19,248, suggesting new E3 ligases may offer great opportunities for degrading novel targets in the proteome through PPI. (page 12)

For example, U2 Small Nuclear RNA Auxiliary Factor 1 (U2AF1), a spliceosome gene related to cancers and myelodysplastic syndrome^{41,42}, has no evidence of PPI with any co-opted E3 ligase but interacts with SMURF1, which may serve as a novel E3 ligase in PROTAC (Figure 5b). (page 12)

Reviewer #2 (Remarks to the Author):

In this manuscript, titled “Charting the Expanding PROTACtable Genome Universe of E3 Ligases,” Liu et al. present a purely computational analysis of E3 ligases at a genome-wide scale, exploring different aspects of them that, intuitively and also based on the relatively limited existing evidence, would qualify as desirable traits for TPD. While, according to the authors' assessment, hundreds of E3 ligases could be “interesting” by at least one of the metrics, they manage to narrow down the search to a list of 70+ E3s that are worth exploring further. There is no strong validation of any of the newly identified E3 ligases, but this is arguably not the goal of the work; rather, the authors chose to provide a panoramic view of the E3 ligase space, quantifying their ligandability, localization in cellular compartments, expression across tissues, and PPIs, among others. This is a commendable effort, and I appreciate the fact that all of this information is available through an intuitive web interface. I am convinced that this resource will be of interest to the community. Another aspect I appreciate is the focus on cancer and the widespread analysis in this context.

Response: We thank the reviewer for his overall positive evaluation of our work.

In my opinion, the weakest point of the paper is the assessment of ligandability of E3 ligases. Although, broadly speaking, the choice of resources makes sense (DrugBank/DGIdb for drugs and “well-known” interactions, ChEMBL for ligand-protein interactions, and electrophiles from a recent study), the authors should acknowledge that some of the resources are highly knowledge-biased, which yields trivial observations, such as high ligandability for highly interesting actors such as VHL or CRBN. This is difficult to mitigate, but the authors should make this more explicit early in the text. Also, I would highly recommend that categories such as Pharos (Tdark, Tbio, Tchem, Tclin) are explored.

Response: We thank the reviewer for these valuable suggestions. We fully acknowledge that these resources might be influenced by existing knowledge biases. Therefore, we've sourced data from multiple repositories, enabling users to select their preferred data resource. We have revised the introduction of our manuscript, stating that “Current data resources may reflect known biases. Thus, we've incorporated numerous options for users to select data sources based on their preferences.” (page 4)

Per the reviewer suggestion, we assigned the E3 ligases into E3drug, E3chem, and E3cova. E3 ligases with known evidence of interacting with drugs were assigned to E3drug group; E3 ligases with evidence of interacting with small molecules in ChEMBL were assigned to E3chem; and the rest E3 ligases were assigned to E3cova. We updated this in the revised manuscript, stating that “Inspired by Pharos, a druggable genome resource⁴³, we labeled E3 ligases into E3drug, E3chem, E3cova, and E3dark (see Methods). Among the identified E3 ligases, 127 (11.8%) were labeled as E3drug, 75 (7.0%) as E3chem, and 484 (45.0%) as E3cova. Additionally, we quantified the number of ligand categories associated with the E3 ligases.” (page 6), as well as Figure 2a (see below).

Likewise, it is possible that the prediction algorithm (virtual screening; VS) used produces many false positives and, overall, what I think is relevant from the analysis are the aggregate numbers, not the specific ligand-E3 ligase interactions discovered. In a VS exercise like this one, it is important to do leave-ligands-out and leave-proteins-out cross-validations, and I would advise that the authors investigate if, in their context, the reported accuracies apply in stringent cross-validation settings.

Response: Thanks for these valuable suggestions. We added these three cross-validation tests to reflect the different evaluation settings. The four tests we have is: 1) both drug and protein in the testing set could appear in the training sets. 2) drugs in the test set don't appear in the training set. (3) proteins in the test set don't appear in the training set. (4) both drugs and proteins in the test set don't appear in the training set. The 4 experimental settings cover leave-ligands-out and leave-protein-out, and the stricter test that neither the drug nor the protein has been seen during training. As expected, the AUC decreases when the evaluation becomes more stringent. The aggregate number from predictions is significantly influenced by criteria chosen. We've uploaded our prediction results to our web portal. Users can specify the top results and determine cutoff values, allowing them to view the aggregate figures directly on our web portal.

We have revised the figure S3 (see below) and manuscript as follows: “To search for the potentially new drugs for recruiting E3 ligases, we adopted a deep-learning based virtual screening ligand model, HyperAttentionDTI¹⁷, to search for potential interactions between E3 ligases and drugs in DrugBank¹⁸. The model captured complex interactions between atoms and amino acids through an innovative attention mechanism and achieved improved performance over the state-of-the-art baselines, and the new drug-target interactions falls into top virtual screening predicted interactions in the test case study. After evaluating and applying this model (Figure S3), we obtained predicted drug-target interactions, including for potentially novel E3 ligases, such as Makorin Ring Finger Protein 1 (MKRN1), having PPIs with key tumor drivers, TP53 and APC32, and predicted to interact with DB13955 (Estradiol dianthate) and DB03017 (Lauric acid). Utilizing these drugs may increase the ligandability of potentially novel E3 ligases that target additional tumor targets. Taken together, our systematic analysis exploited five large-scale data sources and revealed a number of new E3 ligases with tens of available ligands that may be utilized for PROTAC development.” (page 7). We also added the pertaining text in the Methods as follows: “We performed the evaluation employing four configurations: 1) both drugs and proteins in the

test set are present in the training sets; 2) drugs in the test set are excluded from the training set; 3) proteins in the test set are excluded from the training set; and 4) both drugs and proteins in the test set are absent in the training set. These setups range from traditional leave-ligands-out and leave-protein-out methods to a stringent test ensuring that neither drug nor protein was seen during training.” (page 22).

Figure S3. The evaluation of in silico prediction model of drug-E3 ligase interactions. (a) AUC-ROC curve of HyperAttentionDTI in the cross-validation that both drugs and proteins in the testing set appear in the training set (b) AUC-ROC curve that drugs in the testing set don't appear in the training set. (c) AUC-ROC curve that proteins in the testing set don't appear in the training set. (d) AUC-ROC curve that both drugs and proteins in the testing set don't appear in the training set.

Apart from this, I think the work is sound and provides an interesting resource to the community. There is an enormous amount of literature discussing TPD and the role of E3 ligases, so I think the discussion could be richer. In particular, if so many E3 ligases appear to be interesting, why are only a few progressed? What makes it so difficult to go beyond VHL, CRBN, and the usual suspects? The authors could comment on that. Given the high interest in TPD, it is highly unlikely that a majority of E3 ligases have not been already explored as solid candidates: the authors should conduct an in-depth literature search, at least for their 70+ candidates, and see if some of them already have interesting chemotypes that are being investigated by medicinal chemists. Otherwise, the paper leaves the impression that most E3 ligases are mostly underexplored, which is certainly not true.

Response: We thank the reviewer for this valuable suggestion. Despite E3 ligases' pivotal role in protein homeostasis being well-documented, their exploration within the context of targeted

protein degradation (TPD), particularly PROTACs, has been somewhat limited. The majority of developed PROTACs predominantly utilize either VHL or CRBN, as evidenced by PROTACpedia (<https://protacpedia.weizmann.ac.il>), a resource actively collating PROTACs, where 94.3% (766/812) of PROTACs recruited either of these two ligases.

The field of PROTAC itself is relatively nascent, with the first translational research starting only in 2019. Since then, the community has been actively investigating the target scope of PROTACs³. The potency of PROTACs can be significantly affected by numerous factors including the choice of POI binder, linker length, and linker type, where even minor alterations can drastically impact degradation potency⁴⁴. As such, much of the initial efforts have been devoted to pairing VHL or CRBN with different target proteins, in a bid to save resources and expedite the progress.

However, the development of resistance to PROTACs and the increasing demand for precision in TPD have underscored the need for additional E3 ligases³. This has motivated us to conduct our study, aiming to assist researchers in developing novel E3 ligases to expand the usage of PROTACs. This, in turn, underscores the significance of our work and the potential it holds for the field. We discussed this in the introduction “The PROTAC field is still in its early stages, with the initial translational research commencing in 2019. Since then, the community has diligently explored the target scope of PROTACs³, but there has been comparatively less effort dedicated to searching new E3 ligases. Several factors, such as ligandability, expression, cellular localization, and many others, can influence the efficacy of a PROTAC. These numerous factors make the search for new E3 ligases challenging. Much of the initial efforts have been devoted to pairing VHL or CRBN with different target proteins in order to save resources and expedite the progress.” (page 3) as well as now in the discussion section “Our study built upon insights from prior research to identify new E3 ligases that could be integrated into PROTAC development. As future experiments unveil more insights, we could identify additional factors and resources to provide more precise guidance on the search for new E3 ligases.” (page 20) in the revised manuscript.

In addition, I see in the figures proteins like AURKA which are not bona fide E3 ligases. I would recommend that, in the figures, the authors use their literature confidence score to filter out cases like this one. I think it is important that all E3 ligases appearing in the figures are broadly accepted to perform this function in the cell.

Response: Thanks for the valuable suggestions. Regarding the E3 ligase functions of AURKA, the gene was collected in Ge et al. 2018 and UbiBrowser as E3 ligase. AURKA was considered as E3 ligase in Ge et al. based on literature³⁴, and in UbiBrowser based on literatures³⁵. The evidence suggests that AURKA is very likely to function as an E3 ligase. More generally, we have gathered evidence from the credible sources, suggesting that certain genes could function as E3 ligases. We prefer not to engage in debates regarding the authenticity of a gene as a bona fide E3 ligase and have thus chosen to avoid claiming “bona fide E3 ligases” in our manuscript. We removed the highlighted description of AURKA in the text of the revised manuscript. We will be happy to further discuss if the reviewer has any remaining concerns.

Finally, I would recommend that authors revise wording and terminology. For example:

- The title: “charting the expanding” is a bit confusing in the title. Wouldn't one of the two words be enough?

Response: We revised it to: “Expanding PROTACtable genome universe of E3 ligases”.

- POI usually means “Protein of Interest”, not “Point of Interest protein”

Response: We revised throughout the manuscript to indicate that POI stands for "Protein of Interest"

- In page 5, they refer to ligands from different sources as "types of ligands". The source is not a proper way to classify ligands by "type".

Response: We changed all ‘type’ into ‘source’ in the revised manuscript.

We have revised our manuscript as “We identified 686 (63.8%) E3 ligases that interact with known ligands from at least one category of ligand sources (Figure 2a & Figure S2a). Specifically, 127 (11.8%) E3 ligases had evidence of targeting by or interacting with drugs in DrugBank25 and/or DGIdb26, 145 (13.5%) E3 ligases have bioactive ligands in ChEMBL, and 626 (58.2%) can interact with electrophiles in SLCABPP (Figure 2a & Figure S2b). Inspired by Pharos, a druggable genome resource²⁹, we labeled E3 ligases into E3drug, E3chem, E3cova, and E3dark (see Methods). Among the identified E3 ligases, 127 (11.8%) were labeled as E3drug, 75 (7.0%) as E3chem, and 484 (45.0%) as E3cova. Additionally, we quantified the number of ligand sources associated with the E3 ligases. We identified that 55 (5.2%) E3 ligases can interact with all three sources of ligands (Figure 2a & Figure S2a), including the clinically used VHL and CRBN, and the experimentally explored KEAP1, XIAP, MDM2, BIRC2, and AHR. Beyond these E3 ligases, 48 have not yet been reported to be co-opted in PROTAC, pointing them as potential candidates for expanding E3 ligases for PROTACs. For instance, Tripartite-motif protein 24 (TRIM24)³⁰ has

evidence of available ligands from drug, ChEMBL, and SLCABPP (Figure 2b), example ligands including salicylaldehyde (ChEMBL108925) and the inhibitor (DGIdb: 252166607). In addition to the count of ligand sources, we also quantified the total number of ligands per E3 ligase. 77 (7.2%) E3 ligases have over 300 ligands, including 7 out of 12 co-opted E3 ligases (Figure 2c).”(page 7)

Reviewer #3 (Remarks to the Author):

This article presents an integration of comprehensive data from multiple resources to investigate the activity of E3 ligases in human cells as a key factor in target protein degradation. The study pipeline is well-established, and the data sources are comprehensive. Additionally, the authors have created a user-friendly web portal to search for the data and present the results of the study.

Response: We thank the reviewer for her/his positive evaluation of our work.

However, before publishing, there are some minor revisions that should be considered:

1. In the expression data, the authors should provide clarification on all pre-processing analyses, such as the removal of genes with very low expression, identification and removal of possible batch effects, and identification and removal of outlier samples.

Response: Thanks. Given the large volume of omics data involved, we have taken great care in selecting our data sources to ensure quality. We opted to utilize the UCSC Toil RNAseq Recompute Compendium for bulk transcriptomics expression data derived from TCGA and GTEx⁴⁵. This compendium processes both TCGA and GTEx using a unified pipeline and includes quality control measures and batch effect mitigation. We have implemented median expression level analysis to minimize outlier effects. For protein expression, we obtained data from the Human Protein Atlas pathology, and binarized samples into high-medium and low-NA groups, following the methodology of a recent study²³.

Regarding the tumor single-cell transcriptomics data, we chose the Tumor Immune Single-cell Hub (TISCH), a scRNA-seq database specializing in the tumor microenvironment (TME)⁴⁶, to examine the expression level of E3 ligases in tumors at the single-cell level. All single-cell datasets in TISCH underwent uniform quality control, clustering, and cell-type annotation. We only utilized those single-cell datasets that originate from humans, are untreated, and are analyzed via 10X Genomics. For single-cell transcriptomics data in normal tissues, we used Tabula Sapiens, a benchmark human cell atlas that provides a preliminary map of nearly 500,000 cells across 24 organs from 15 normal human subjects⁴⁷. Noteworthy features of Tabula Sapiens include rapid sample processing, expert cell annotation, and rigorous quality control. We clarified that the most pre-processing was conducted by the original data sources. For example, TISCH has quantified and minimized the batch effects across datasets⁴⁶.

We added this detailed information in the revised manuscript, in the Method section (page 22 -23).

2. The public PPI datasets contain a high rate of false positive interactions, such as those obtained solely through text mining. The authors should explain how they reduced the false positive rate in the PPI data.

Response: Thanks for the insightful suggestion. We indeed have performed further analysis to reduce the potential false positive rate. For PPI sources obtained through text mining (e.g., BioGrid, IntAct, Reactome), we collected details on interaction types and experimental systems (when available), which are annotated in our web portal. While we have retained all PPIs to offer a

comprehensive dataset, we've marked certain interactions with an asterisk. These marked PPIs, which were not based on physical association or were not identified via methods such as yeast two-hybrid studies, affinity purification-mass spectrometry, protein 3D structures, or low-throughput experiments, serve as indicators for users that there might be a risk of false positives.

Additionally, we have revised the Methods section related to PPIs to provide a clearer understanding of the quality control process implemented for collected PPIs. The pertaining text now reads: “General PPIs between E3 ligase and target protein were collected from BioGrid⁴⁸ (<https://thebiogrid.org/>), IntAct⁴⁹ (<https://www.ebi.ac.uk/intact/>), and Reactome⁵⁰ (<https://reactome.org/>), HuRI⁵¹ (<http://www.interactome-atlas.org/>), STRING⁵² (<https://string-db.org/>). To ensure the quality of the incorporated PPIs, we extracted PPIs from literature curation sources such as BioGrid, IntAct, and Reactome, carefully removing any non-human entries. We marked PPIs that were in physical association identified via methods such as yeast two-hybrid studies, affinity purification-mass spectrometry, protein 3D structures, or low-throughput experiments, serving as indicators for users that these PPIs were obtained in a high confidence. Entries from HuRI were retained due to their comprising solely of experimentally verified human binary protein interactions. PPIs from STRING were combined with known and predicted PPIs, and we only retained those human-related and with high confidence (score >700). PPIs specific for E3-substrate interactions (ESIs) were retrieved from from UbiBrowser2.0²⁹ (<http://ubibrowser.bio-it.cn/>), retaining only human ESIs documented in literature and high-confidence predicted ESIs with scores exceeding 0.7.”(page 24)

Target	Detection Methods	PPI type	Source	Interface E3	Interface Target	High Confidence*
ABTB2	Affinity Capture-MS	physical	BioGrid	-	-	Yes
ACTR6	Affinity Capture-MS	physical	BioGrid	-	-	Yes
AFF4	Affinity Capture-MS	physical	BioGrid	-	-	Yes
AHSG	Affinity Capture-MS	physical	BioGrid	-	-	Yes
ALDOA	Affinity Capture-MS	physical	BioGrid	-	-	Yes
ANXA5	Affinity Capture-MS	physical	BioGrid	-	-	Yes

3.It would be interesting to conduct an enrichment analysis on the E3 ligase targets to investigate the biological processes and pathways involved and their differences in normal and tumour samples.

Response: Thanks for this valuable suggestion. We agree that the enrichment analysis might uncover the diverse functions of the gene. For instance, we performed Gene Ontology (GO) and KEGG pathway enrichment analysis on potential targets of TRIM24. According to the UniProt gene function annotation (<https://www.uniprot.org/uniprotkb/O15164/entry>), TRIM24 functions include transcriptional control, E3 ligase activity mediating TP53 degradation, among others. Both GO and KEGG enrichment analyses underscore the role of transcription regulation, with the KEGG enrichment analysis indicating enrichment in multiple cancer pathways.

Figure. GO enrichment analysis of potential targets of TRIM24

Figure. KEGG enrichment analysis of potential targets of TRIM24

Another example pertains to the enrichment analysis conducted on proteins interacting with CRBN. As per the UniProt gene function annotation (<https://www.uniprot.org/uniprotkb/Q96SW2/entry>), CRBN's functions encompass E3 ligase activity, regulation of large-conductance calcium-activated potassium channels, among others. The pathways enriched in this context include those involved with transporters and calcium signaling pathways, among others.

Figure. GO enrichment analysis of potential targets of CRBN

Figure. KEGG enrichment analysis of potential targets of CRBN

As evident in these examples, the enriched pathways are highly diverse and closely tied to the endogenous functions of these proteins. Therefore, we have incorporated GO and KEGG

enrichment analysis into our web portal to cater to researchers' interests (see screenshot below). We added this in our revised manuscript, stating that “Recognizing that the PPI of E3 ligases might provide insights into the gene's inherent functions, our web portal incorporates both Gene Ontology and KEGG enrichment analyses on proteins interacting with each E3 ligase.” (page 16).

It's important to note that it is challenging to determine whether targets of E3 ligases are generally enriched in tumor or normal tissues, as this can significantly vary depending on the specific E3 ligase and the biological context, that is the tumor type. We assume that such enrichment analysis are out of scope of the current work and could be performed interpedently by readers according to their specific interests.

PPI

GO Enrichment KEGG Enrichment

Show entries Search:

Target	Detection Methods	PPI type	Source	Interface E3	Interface Target	High Confidence*
ABTB2	Affinity Capture-MS	physical	BioGrid	-	-	Yes
ACTR6	Affinity Capture-MS	physical	BioGrid	-	-	Yes
AFF4	Affinity Capture-MS	physical	BioGrid	-	-	Yes
AHSG	Affinity Capture-MS	physical	BioGrid	-	-	Yes

View enrichment analysis result

4. Investigating related pathways for E3 ligase targets or for extended PPIs (indirect targets) could be useful in identifying any possible side effects of specific protein degradation.

Response: We thank for the suggestion. We added targets enriched pathways in the data portal (please refer to above comment). A single E3 ligase may interact with hundreds/thousands of proteins, and each of these proteins could further interact with hundreds/thousands more. For example, in the high confidence STRING human dataset, 298 unique proteins interact with VHL, leading to a total of 98,259 unique extended PPIs. Given the current lack of empirical evidence on how to further reliably rank or filter these interactions, we find that this extensive network of extended PPIs would likely be too complex for readers to navigate and understand (or, for this matter, to ourselves, to be honest...). Therefore, the possible inclusion of extended PPI information should in our minds be best left for a future study.

Reference

1. Bondeson, D. P. et al. Lessons in PROTAC Design from Selective Degradation with a Promiscuous Warhead. *Cell Chemical Biology* 25, 78-87.e5 (2018).
2. Schneider, M. et al. The PROTACtable genome. *Nat Rev Drug Discov* (2021) doi:10.1038/s41573-021-00245-x.
3. Békés, M., Langley, D. R. & Crews, C. M. PROTAC targeted protein degraders: the past is prologue. *Nat Rev Drug Discov* 1–20 (2022) doi:10.1038/s41573-021-00371-6.
4. Alexandrov, L. B. et al. The repertoire of mutational signatures in human cancer. *Nature* 578, 94–101 (2020).
5. Calabrese, C. et al. Genomic basis for RNA alterations in cancer. *Nature* 578, 129–136 (2020).
6. Yuan, Y. et al. Comprehensive molecular characterization of mitochondrial genomes in human cancers. *Nat Genet* 52, 342–352 (2020).
7. Rodriguez-Martin, B. et al. Pan-cancer analysis of whole genomes identifies driver rearrangements promoted by LINE-1 retrotransposition. *Nat Genet* 52, 306–319 (2020).
8. Hoadley, K. A. et al. Cell-of-Origin Patterns Dominate the Molecular Classification of 10,000 Tumors from 33 Types of Cancer. *Cell* 173, 291-304.e6 (2018).
9. Chen, H. et al. A Pan-Cancer Analysis of Enhancer Expression in Nearly 9000 Patient Samples. *Cell* 173, 386-399.e12 (2018).
10. Chiu, H.-S. et al. Pan-Cancer Analysis of lncRNA Regulation Supports Their Targeting of Cancer Genes in Each Tumor Context. *Cell Reports* 23, 297-312.e12 (2018).
11. Berger, A. C. et al. A Comprehensive Pan-Cancer Molecular Study of Gynecologic and Breast Cancers. *Cancer Cell* 33, 690-705.e9 (2018).
12. Gong, J. et al. A Pan-cancer Analysis of the Expression and Clinical Relevance of Small Nucleolar RNAs in Human Cancer. *Cell Rep* 21, 1968–1981 (2017).
13. Ye, Y. et al. The Genomic Landscape and Pharmacogenomic Interactions of Clock Genes in Cancer Chronotherapy. *Cell Syst* 6, 314-328.e2 (2018).
14. Zhang, Z. et al. Transcriptional landscape and clinical utility of enhancer RNAs for eRNA-targeted therapy in cancer. *Nat Commun* 10, 4562 (2019).
15. Ye, Y. et al. Sex-associated molecular differences for cancer immunotherapy. *Nat Commun* 11, 1779 (2020).
16. Ye, Y. et al. Characterization of Hypoxia-associated Molecular Features to Aid Hypoxia-Targeted Therapy. *Nat Metab* 1, 431–444 (2019).
17. Zhao, Q., Zhao, H., Zheng, K. & Wang, J. HyperAttentionDTI: improving drug–protein interaction prediction by sequence-based deep learning with attention mechanism. *Bioinformatics* 38, 655–662 (2022).
18. Wishart, D. S. et al. DrugBank 5.0: a major update to the DrugBank database for 2018. *Nucleic Acids Research* 46, D1074–D1082 (2018).
19. He, M. et al. PROTACs: great opportunities for academia and industry (an update from 2020 to 2021). *Sig Transduct Target Ther* 7, 1–64 (2022).
20. Moreau, K. et al. Proteolysis-targeting chimeras in drug development: A safety perspective. *Br J Pharmacol* 177, 1709–1718 (2020).
21. Chen, C. et al. Recent Advances in Pro-PROTAC Development to Address On-Target Off-Tumor Toxicity. *J. Med. Chem.* 66, 8428–8440 (2023).
22. Khan, S. et al. A selective BCL-XL PROTAC degrader achieves safe and potent antitumor activity. *Nat Med* 25, 1938–1947 (2019).

23. MacKay, M. et al. The therapeutic landscape for cells engineered with chimeric antigen receptors. *Nat Biotechnol* 38, 233–244 (2020).
24. Jing, Y. et al. Expression of chimeric antigen receptor therapy targets detected by single-cell sequencing of normal cells may contribute to off-tumor toxicity. *Cancer Cell* 39, 1558–1559 (2021).
25. Jing, Y. et al. Association Between Sex and Immune-Related Adverse Events During Immune Checkpoint Inhibitor Therapy. *JNCI: Journal of the National Cancer Institute* djab035 (2021) doi:10.1093/jnci/djab035.
26. Jing, Y., Yang, J., Johnson, D. B., Moslehi, J. J. & Han, L. Harnessing big data to characterize immune-related adverse events. *Nat Rev Clin Oncol* 19, 269–280 (2022).
27. Ge, Z. et al. Integrated Genomic Analysis of the Ubiquitin Pathway across Cancer Types. *Cell Reports* 23, 213–226.e3 (2018).
28. Liu, L. et al. UbiHub: a data hub for the explorers of ubiquitination pathways. *Bioinformatics* 35, 2882–2884 (2019).
29. Wang, X. et al. UbiBrowser 2.0: a comprehensive resource for proteome-wide known and predicted ubiquitin ligase/deubiquitinase–substrate interactions in eukaryotic species. *Nucleic Acids Research* (2021) doi:10.1093/nar/gkab962.
30. Ge, K., X, Z. & Ze, F. PPAR- γ AF-2 domain functions as a component of a ubiquitin-dependent degradation signal. *Obesity (Silver Spring, Md.)* 17, (2009).
31. Hou, Y. et al. PPAR γ E3 ubiquitin ligase regulates MUC1-C oncoprotein stability. *Oncogene* 33, 5619–5625 (2014).
32. Lee, J. H. et al. Degradation of selenoprotein S and selenoprotein K through PPAR γ -mediated ubiquitination is required for adipocyte differentiation. *Cell Death Differ* 26, 1007–1023 (2019).
33. Hou, Y., Moreau, F. & Chadee, K. PPAR γ is an E3 ligase that induces the degradation of NF κ B/p65. *Nat Commun* 3, 1300 (2012).
34. Briassouli, P., Chan, F. & Linardopoulos, S. The N-terminal domain of the Aurora-A Phe-31 variant encodes an E3 ubiquitin ligase and mediates ubiquitination of IkappaBalpha. *Hum Mol Genet* 15, 3343–3350 (2006).
35. Yang, C. et al. Effects of AURKA-mediated degradation of SOD2 on mitochondrial dysfunction and cartilage homeostasis in osteoarthritis. *J Cell Physiol* 234, 17727–17738 (2019).
36. Dale, B. et al. Advancing targeted protein degradation for cancer therapy. *Nat Rev Cancer* (2021) doi:10.1038/s41568-021-00365-x.
37. Hines, J., Lartigue, S., Dong, H., Qian, Y. & Crews, C. M. MDM2-Recruiting PROTAC Offers Superior, Synergistic Antiproliferative Activity via Simultaneous Degradation of BRD4 and Stabilization of p53. *Cancer Research* 79, 251–262 (2019).
38. Marcellino, B. et al. Development of an MDM2 Degradator for Treatment of Acute Leukemias. *Blood* 138, 1866 (2021).
39. Ohoka, N. et al. Development of Small Molecule Chimeras That Recruit AhR E3 Ligase to Target Proteins. *ACS Chem. Biol.* 14, 2822–2832 (2019).
40. Zhang, X. et al. Discovery of IAP-recruiting BCL-XL PROTACs as potent degraders across multiple cancer cell lines. *European Journal of Medicinal Chemistry* 199, 112397 (2020).
41. Yoshida, H. et al. Elucidation of the aberrant 3' splice site selection by cancer-associated mutations on the U2AF1. *Nat Commun* 11, 4744 (2020).
42. Cheruiyot, A. et al. Nonsense-Mediated RNA Decay Is a Unique Vulnerability of Cancer Cells Harboring SF3B1 or U2AF1 Mutations. *Cancer Research* 81, 4499–4513 (2021).

43. Nguyen, D.-T. et al. Pharos: Collating protein information to shed light on the druggable genome. *Nucleic Acids Research* 45, D995–D1002 (2017).
44. Schapira, M., Calabrese, M. F., Bullock, A. N. & Crews, C. M. Targeted protein degradation: expanding the toolbox. *Nat Rev Drug Discov* 18, 949–963 (2019).
45. Vivian, J. et al. Toil enables reproducible, open source, big biomedical data analyses. *Nat Biotechnol* 35, 314–316 (2017).
46. Sun, D. et al. TISCH: a comprehensive web resource enabling interactive single-cell transcriptome visualization of tumor microenvironment. *Nucleic Acids Research* 49, D1420–D1430 (2021).
47. Consortium, T. T. S. & Quake, S. R. The Tabula Sapiens: a multiple organ single cell transcriptomic atlas of humans. 2021.07.19.452956 Preprint at <https://doi.org/10.1101/2021.07.19.452956> (2021).
48. Oughtred, R. et al. The BioGRID interaction database: 2019 update. *Nucleic Acids Research* 47, D529–D541 (2019).
49. Orchard, S. et al. The MIntAct project—IntAct as a common curation platform for 11 molecular interaction databases. *Nucleic Acids Research* 42, D358–D363 (2014).
50. Gillespie, M. et al. The reactome pathway knowledgebase 2022. *Nucleic Acids Research* 50, D687–D692 (2022).
51. Luck, K. et al. A reference map of the human binary protein interactome. *Nature* 580, 402–408 (2020).
52. Szklarczyk, D. et al. STRING v10: protein-protein interaction networks, integrated over the tree of life. *Nucleic Acids Res* 43, D447–452 (2015).

Re: NCOMMS-23-12374-T

REVIEWER COMMENTS

Reviewer #1 (Remarks to the Author):

Liu et al describe a project in which data on E3 ligases from various sources are integrated, with a view to assisting the selection of new/novel members of that family for example in PROTAC development. The integrated information is made available via a public website. This work would be of interest to the broader protein degradation community and is worthy of publication. However, I don't consider it to contain sufficient "noteworthy" original results to warrant publication in Nature Communications. In some respects it is an incremental advance on previous work such as the PROTACtable Genome analysis of Schneider et al; furthermore the paper does not provide any particularly insightful analyses of the E3 ligase landscape beyond some illustrative examples throughout the manuscript (none of which have any experimental validation). It would seem to me that a more suitable home for such a study would be the companion Nature journal Scientific Data (though this is ultimately a decision for the editor).

Response: PROTACs employ E3 ligases to tag target proteins for subsequent degradation, and the recruitment of E3 ligases is an essential step in the process of target protein degradation¹. The PROTACtable Genome analysis of Schneider et al is indeed a comprehensive and very worthy analysis focusing on the target proteins of PROTACs², but it did not consider the potential of new E3 ligase that could be co-opted into the PROTAC development. Furthermore, the prevalent use of VHL and CRBN has presented a challenge to the current development of PROTACs³, thereby motivating our search for novel potential E3 ligases. Please note, our work is the first comprehensive analysis focusing on the features of E3 ligases in the PROTAC, and in this respect is clearly not an incremental advance compared to previous work, e.g., PROTACtable genome analysis.

To further explicate why this is indeed the case, we note that our study has carried out a comprehensive evaluation of all E3 ligases: to this end we collected 30 datasets covering different aspects of PROTAC, including chemical ligandability, expression patterns, protein-protein interactions (PPI), structure availability, functional essentiality, cellular location, and PPI interface. We then systematized and ranked the E3 ligases that could be prioritized into PROTAC development. We further provide a few examples to illustrate the methodology of identifying potent E3 ligases that could act against key cancer targets. Finally, we developed a user-friendly flexible web portal (<https://hanlaboratory.com/E3Atlas>) that is aimed at assisting the broader protein degradation community to rapidly identify new E3 ligases with promising TPD activities against specifically desired targets, facilitating the development of PROTAC in cancer and beyond.

An important novelty of our work relates to the computational-centralized focus. Our approach follows recent studies that have established the significance of rigorous analysis which is based on large-scale data. For example, the recent TCGA/ICGC Pan-cancer project published a series of high-profile papers in Nature and Cell series journals⁴⁻¹¹. These comprehensive analyses provide the foundation and important insights for further investigations, which may be performed

independent of experimental validation (as demonstrated in studies from our group in the cancer genomics field¹²⁻¹⁶).

I do have serious concerns about the inclusion of output from predictive models alongside data from ‘real’ (often peer-reviewed) experiments. The ‘hyperattentionDTI’ model in particular in my view provides results that are of no real value and indeed may well be misleading. An average of 552 drug-ligase interactions is frankly unbelievable given the typical number of meaningful protein:ligand interactions of marketed drugs for example from chemoproteomics experiments. I would advise the authors to seek out someone with expertise in the practical development, validation and use of in silico models in drug discovery (preferably from an industrial background) who will be able to advise on how best to utilise such methods.

Response: We thank for this valuable comment. We reevaluated our method, and adopted the data presentation format from the original HyperAttentionDTI study, which highlights predicted interactions with the highest probability as potential candidate interactions¹⁷. Addressing your concern, in the revised manuscript, we now present only the top predicted drug-target interactions for each E3 ligase. By default, the top 10 predicted interactions are displayed, and users have the flexibility to expand this list from 1 to 500 at their choice. Furthermore, we also included an option that allows users to choose a threshold to examine predicted interactions meeting a specific cutoff.

The Screenshot of virtual screening of data portal has been modified accordingly:

Virtual Screening - HyperAttention

Top prediction: 1 ● 500 Top 10 predictions & Probability: 0.5 ● 1 Probability ≥ 0.9

Show 5 entries

Adjustable range slider of number top predictions

Adjustable range slider of probability cut-off

Identifier	Label	E3 Ligase	Probability (0-1, 1 highest)
DB11327	Yes	TRIM24	1
DB00812	Yes	TRIM24	1
DB04400	Yes	TRIM24	0.9949
DB00991	Yes	TRIM24	0.9939
DB12010	Yes	TRIM24	0.9913

Showing 1 to 5 of 10 entries

Previous 1 2 Next

Predictions descent ordered by probability

We revised the main text accordingly, which now reads (page 7): “To search for the potentially new drugs for recruiting E3 ligases, we adopted a deep-learning based virtual screening ligand model, HyperAttentionDTI¹⁷, to search for potential interactions between E3 ligases and drugs in DrugBank¹⁸. The model captured complex interactions between atoms and amino acids through an innovative attention mechanism and achieved improved performance over the state-of-the-art baselines, and the new drug-target interactions falls into top virtual screening predicted interactions in the test case study. After evaluating and applying this model (Figure S3), we obtained predicted drug-target interactions, including for potentially novel E3 ligases, such as Makorin Ring Finger Protein 1 (MKRN1), having PPIs with key tumor drivers, TP53 and APC32, and predicted to

interact with DB13955 (Estradiol dienanthate) and DB03017 (Lauric acid). Utilizing these drugs may increase the ligandability of potentially novel E3 ligases that target additional tumor targets. Taken together, our systematic analysis exploited five large-scale data sources and revealed a number of new E3 ligases with tens of available ligands that may be utilized for PROTAC development.”.

The authors seem to be in two minds as to whether they are addressing a cancer audience (which would be entirely reasonable given their own areas of research focus) or a wider, disease-agnostic audience (as implied in the Introduction).

Response: Thanks. Considering the wide use of PROTAC in many areas, such as cancer, immune disorder, and neurodegenerative diseases¹⁹, we aimed at providing insights for broader fields beyond cancer. Accordingly, many of E3 ligases properties carefully evaluated in this manuscript are general characteristics involved in PROTAC development. We clarified this in the revised manuscript (page 18-19), stating that “Beyond facilitating the development of PROTACs targeting cancer, aspects such as confidence scores, ligandability, PPI, structural information, cellular localization, and PPI interfaces of E3 ligases also hold broad applicability in the deployment of PROTACs for a range of other diseases. These factors, therefore, present universal value in the context of PROTAC-oriented therapeutic strategies.”

Some additional caveats concerning the real-world relevance of certain data types may also be in order and worthy of comment or more in-depth analysis. For example, how high in reality is the risk of on-target off-tumour tox with clinical degraders? Especially as, by design, PROTACs are not necessarily very potent inhibitors at the relevant E3 ligase.

Response: We thank for this insightful comment. PROTACs can effectively suppress protein activity, and on-target toxicity has become one of the major concerns when these target proteins are essential in normal tissues^{20,21}. To mitigate the on-target toxicity of PROTAC, one efficient method is to recruit E3 ligases that are selectively expressed in desired cell types or tissues. As a result, the PROTAC is less likely to exert its function in the undesired cells or tissues²¹. For example, when developing BCL-XL-targeting PROTACs, VHL was deliberately selected to relieve the on-target toxicity due to VHL's unique expression pattern²². Novel advanced cancer therapeutics, such as CAR-T therapy, were reported to have serious and potentially life-threatening toxicities when more products move into clinical trials^{23,24}. With the first PROTAC entering clinical trials in 2019 and more on the way, we expect more cases of toxicity will be reported. Thereby, our works has firstly tried to establish a systematic platform in order to circumvent on-target, off-tumor toxicity and recommend close monitoring on potential signs of toxicity. We clarified this in the revised manuscript as follows: “PROTACs effectively suppress protein activity, but when they target essential proteins, on-target toxicity may arise^{20,21}. To minimize this, PROTACs can be designed to interact with E3 ligases that are selectively expressed in specific tissues or cell-types, mitigating undesired effects²¹. As an example, VHL's unique expression was leveraged in BCL-XL-targeting PROTACs to reduce toxicity²². As other novel treatments like immunotherapy may lead to serious toxicity²³⁻²⁶, and with the first PROTAC entering clinical tests in 2019, we expect more toxicity reports. Thus, we developed a platform to evaluate such potential on-target, off-tumor effects and aid in toxicity monitoring.” (page 20).

I found the specific references to PPARG and AURKA rather odd; are these really confirmed E3 ligases? Furthermore, when I tested the website using these two proteins the system appeared to 'hang' and provided no results, which did not inspire much confidence.

Response: Collecting a list of E3 ligase is challenging due to the diverse and integrative nature of the ubiquitin pathway²⁷. We therefore assembled credible E3 ligase lists from three different sources, including UbiHub²⁸, Ge et al.²⁷, and UbiBrowser2.0²⁹, and assigned a confidence score (1-6). This was done to capture a comprehensive list of candidate E3 ligases as is currently best possible.

Regarding the E3 ligase functions of two specific genes, PPARG and AURKA, both genes were collected in Ge et al. 2018 and UbiBrowser as E3 ligases. PPARG was considered as E3 ligases in Ge et al. based on literature³⁰, and in UbiBrowser based on literatures³¹⁻³³. AURKA was considered as E3 ligase in Ge et al. based on literature³⁴, and in UbiBrowser based on literatures³⁵. This evidence suggests that PPARG and AURKA are very likely function as an E3 ligase.

Regarding to the display issues, we apologize for that and have carefully resolved the issue and any other remaining issues, to the best of our knowledge.

That aside, the website did seem to function as advertised for more established E3 ligases and was straightforward to use. I would however like to see an option to download the entire dataset for exploration using a user's own data analysis tools.

Response: Thanks. We have now added a download tab to allow users process the data at their own preference.

Screenshot of download page:

E3 Ligase Landscape Home E3 Profile General Search Search by Target Document Download Contact

Download

Given the intricacies of data structure and the necessity for efficient network transmission, the data is housed in the Feather format. Feather is a highly efficient, fast columnar data format compatible with several programming languages, including Python and R.

E3 ligase general information

Expression

- E3 ligase - TCGA
- E3 ligase - GTEx
- E3 ligase - HPA pathology
- E3 ligase - Tabula
- E3 ligase - TISCH

HyperAttentionDTI

- E3 ligase - HyperAttentionDTI

PPI

- E3 ligase - PPI

Dependency

- E3 ligase - CRISPR based
- E3 ligase - RNAi based

There were also a few relatively minor typos that will need to be fixed before publication.

Response: We have revised our manuscript and corrected typos in the revised manuscript. For example:

The ideal E3 ligases of PROTAC for cancer therapy will be those highly expressed in tumors but sparingly expressed in normal tissues³⁶. (page 9)

The three co-opted E3 ligases, VHL, XIAP, and AHR, have been explored against cancer targets in PROTAC^{22,37-40}, and their expression patterns in tumor and normal tissues suggest extensive future usage, such as XIAP in breast cancer (Figure 4C & Figure S10). (page 11)

Expanding co-opted E3 ligases to all E3 ligases, the number of targets was increased by 76.1% from 10,930 to 19,248, suggesting new E3 ligases may offer great opportunities for degrading novel targets in the proteome through PPI. (page 12)

For example, U2 Small Nuclear RNA Auxiliary Factor 1 (U2AF1), a spliceosome gene related to cancers and myelodysplastic syndrome^{41,42}, has no evidence of PPI with any co-opted E3 ligase but interacts with SMURF1, which may serve as a novel E3 ligase in PROTAC (Figure 5b). (page 12)

Reviewer #2 (Remarks to the Author):

In this manuscript, titled “Charting the Expanding PROTACtable Genome Universe of E3 Ligases,” Liu et al. present a purely computational analysis of E3 ligases at a genome-wide scale, exploring different aspects of them that, intuitively and also based on the relatively limited existing evidence, would qualify as desirable traits for TPD. While, according to the authors' assessment, hundreds of E3 ligases could be “interesting” by at least one of the metrics, they manage to narrow down the search to a list of 70+ E3s that are worth exploring further. There is no strong validation of any of the newly identified E3 ligases, but this is arguably not the goal of the work; rather, the authors chose to provide a panoramic view of the E3 ligase space, quantifying their ligandability, localization in cellular compartments, expression across tissues, and PPIs, among others. This is a commendable effort, and I appreciate the fact that all of this information is available through an intuitive web interface. I am convinced that this resource will be of interest to the community. Another aspect I appreciate is the focus on cancer and the widespread analysis in this context.

Response: We thank the reviewer for his overall positive evaluation of our work.

In my opinion, the weakest point of the paper is the assessment of ligandability of E3 ligases. Although, broadly speaking, the choice of resources makes sense (DrugBank/DGIdb for drugs and “well-known” interactions, ChEMBL for ligand-protein interactions, and electrophiles from a recent study), the authors should acknowledge that some of the resources are highly knowledge-biased, which yields trivial observations, such as high ligandability for highly interesting actors such as VHL or CRBN. This is difficult to mitigate, but the authors should make this more explicit early in the text. Also, I would highly recommend that categories such as Pharos (Tdark, Tbio, Tchem, Tclin) are explored.

Response: We thank the reviewer for these valuable suggestions. We fully acknowledge that these resources might be influenced by existing knowledge biases. Therefore, we've sourced data from multiple repositories, enabling users to select their preferred data resource. We have revised the introduction of our manuscript, stating that “Current data resources may reflect known biases. Thus, we've incorporated numerous options for users to select data sources based on their preferences.” (page 4)

Per the reviewer suggestion, we assigned the E3 ligases into E3drug, E3chem, and E3cova. E3 ligases with known evidence of interacting with drugs were assigned to E3drug group; E3 ligases with evidence of interacting with small molecules in ChEMBL were assigned to E3chem; and the rest E3 ligases were assigned to E3cova. We updated this in the revised manuscript, stating that “Inspired by Pharos, a druggable genome resource⁴³, we labeled E3 ligases into E3drug, E3chem, E3cova, and E3dark (see Methods). Among the identified E3 ligases, 127 (11.8%) were labeled as E3drug, 75 (7.0%) as E3chem, and 484 (45.0%) as E3cova. Additionally, we quantified the number of ligand categories associated with the E3 ligases.” (page 6), as well as Figure 2a (see below).

Likewise, it is possible that the prediction algorithm (virtual screening; VS) used produces many false positives and, overall, what I think is relevant from the analysis are the aggregate numbers, not the specific ligand-E3 ligase interactions discovered. In a VS exercise like this one, it is important to do leave-ligands-out and leave-proteins-out cross-validations, and I would advise that the authors investigate if, in their context, the reported accuracies apply in stringent cross-validation settings.

Response: Thanks for these valuable suggestions. We added these three cross-validation tests to reflect the different evaluation settings. The four tests we have is: 1) both drug and protein in the testing set could appear in the training sets. 2) drugs in the test set don't appear in the training set. (3) proteins in the test set don't appear in the training set. (4) both drugs and proteins in the test set don't appear in the training set. The 4 experimental settings cover leave-ligands-out and leave-protein-out, and the stricter test that neither the drug nor the protein has been seen during training. As expected, the AUC decreases when the evaluation becomes more stringent. The aggregate number from predictions is significantly influenced by criteria chosen. We've uploaded our prediction results to our web portal. Users can specify the top results and determine cutoff values, allowing them to view the aggregate figures directly on our web portal.

We have revised the figure S3 (see below) and manuscript as follows: “To search for the potentially new drugs for recruiting E3 ligases, we adopted a deep-learning based virtual screening ligand model, HyperAttentionDTI¹⁷, to search for potential interactions between E3 ligases and drugs in DrugBank¹⁸. The model captured complex interactions between atoms and amino acids through an innovative attention mechanism and achieved improved performance over the state-of-the-art baselines, and the new drug-target interactions falls into top virtual screening predicted interactions in the test case study. After evaluating and applying this model (Figure S3), we obtained predicted drug-target interactions, including for potentially novel E3 ligases, such as Makorin Ring Finger Protein 1 (MKRN1), having PPIs with key tumor drivers, TP53 and APC32, and predicted to interact with DB13955 (Estradiol dianthate) and DB03017 (Lauric acid). Utilizing these drugs may increase the ligandability of potentially novel E3 ligases that target additional tumor targets. Taken together, our systematic analysis exploited five large-scale data sources and revealed a number of new E3 ligases with tens of available ligands that may be utilized for PROTAC development.” (page 7). We also added the pertaining text in the Methods as follows: “We performed the evaluation employing four configurations: 1) both drugs and proteins in the

test set are present in the training sets; 2) drugs in the test set are excluded from the training set; 3) proteins in the test set are excluded from the training set; and 4) both drugs and proteins in the test set are absent in the training set. These setups range from traditional leave-ligands-out and leave-protein-out methods to a stringent test ensuring that neither drug nor protein was seen during training.” (page 22).

Figure S3. The evaluation of in silico prediction model of drug-E3 ligase interactions. (a) AUC-ROC curve of HyperAttentionDTI in the cross-validation that both drugs and proteins in the testing set appear in the training set (b) AUC-ROC curve that drugs in the testing set don't appear in the training set. (c) AUC-ROC curve that proteins in the testing set don't appear in the training set. (d) AUC-ROC curve that both drugs and proteins in the testing set don't appear in the training set.

Apart from this, I think the work is sound and provides an interesting resource to the community. There is an enormous amount of literature discussing TPD and the role of E3 ligases, so I think the discussion could be richer. In particular, if so many E3 ligases appear to be interesting, why are only a few progressed? What makes it so difficult to go beyond VHL, CRBN, and the usual suspects? The authors could comment on that. Given the high interest in TPD, it is highly unlikely that a majority of E3 ligases have not been already explored as solid candidates: the authors should conduct an in-depth literature search, at least for their 70+ candidates, and see if some of them already have interesting chemotypes that are being investigated by medicinal chemists. Otherwise, the paper leaves the impression that most E3 ligases are mostly underexplored, which is certainly not true.

Response: We thank the reviewer for this valuable suggestion. Despite E3 ligases' pivotal role in protein homeostasis being well-documented, their exploration within the context of targeted

protein degradation (TPD), particularly PROTACs, has been somewhat limited. The majority of developed PROTACs predominantly utilize either VHL or CRBN, as evidenced by PROTACpedia (<https://protacpedia.weizmann.ac.il>), a resource actively collating PROTACs, where 94.3% (766/812) of PROTACs recruited either of these two ligases.

The field of PROTAC itself is relatively nascent, with the first translational research starting only in 2019. Since then, the community has been actively investigating the target scope of PROTACs³. The potency of PROTACs can be significantly affected by numerous factors including the choice of POI binder, linker length, and linker type, where even minor alterations can drastically impact degradation potency⁴⁴. As such, much of the initial efforts have been devoted to pairing VHL or CRBN with different target proteins, in a bid to save resources and expedite the progress.

However, the development of resistance to PROTACs and the increasing demand for precision in TPD have underscored the need for additional E3 ligases³. This has motivated us to conduct our study, aiming to assist researchers in developing novel E3 ligases to expand the usage of PROTACs. This, in turn, underscores the significance of our work and the potential it holds for the field. We discussed this in the introduction “The PROTAC field is still in its early stages, with the initial translational research commencing in 2019. Since then, the community has diligently explored the target scope of PROTACs³, but there has been comparatively less effort dedicated to searching new E3 ligases. Several factors, such as ligandability, expression, cellular localization, and many others, can influence the efficacy of a PROTAC. These numerous factors make the search for new E3 ligases challenging. Much of the initial efforts have been devoted to pairing VHL or CRBN with different target proteins in order to save resources and expedite the progress.” (page 3) as well as now in the discussion section “Our study built upon insights from prior research to identify new E3 ligases that could be integrated into PROTAC development. As future experiments unveil more insights, we could identify additional factors and resources to provide more precise guidance on the search for new E3 ligases.” (page 20) in the revised manuscript.

In addition, I see in the figures proteins like AURKA which are not bona fide E3 ligases. I would recommend that, in the figures, the authors use their literature confidence score to filter out cases like this one. I think it is important that all E3 ligases appearing in the figures are broadly accepted to perform this function in the cell.

Response: Thanks for the valuable suggestions. Regarding the E3 ligase functions of AURKA, the gene was collected in Ge et al. 2018 and UbiBrowser as E3 ligase. AURKA was considered as E3 ligase in Ge et al. based on literature³⁴, and in UbiBrowser based on literatures³⁵. The evidence suggests that AURKA is very likely to function as an E3 ligase. More generally, we have gathered evidence from the credible sources, suggesting that certain genes could function as E3 ligases. We prefer not to engage in debates regarding the authenticity of a gene as a bona fide E3 ligase and have thus chosen to avoid claiming “bona fide E3 ligases” in our manuscript. We removed the highlighted description of AURKA in the text of the revised manuscript. We will be happy to further discuss if the reviewer has any remaining concerns.

Finally, I would recommend that authors revise wording and terminology. For example:

- The title: “charting the expanding” is a bit confusing in the title. Wouldn't one of the two words be enough?

Response: We revised it to: “Expanding PROTACtable genome universe of E3 ligases”.

- POI usually means “Protein of Interest”, not “Point of Interest protein”

Response: We revised throughout the manuscript to indicate that POI stands for "Protein of Interest"

- In page 5, they refer to ligands from different sources as "types of ligands". The source is not a proper way to classify ligands by "type".

Response: We changed all ‘type’ into ‘source’ in the revised manuscript.

We have revised our manuscript as “We identified 686 (63.8%) E3 ligases that interact with known ligands from at least one category of ligand sources (Figure 2a & Figure S2a). Specifically, 127 (11.8%) E3 ligases had evidence of targeting by or interacting with drugs in DrugBank25 and/or DGIdb26, 145 (13.5%) E3 ligases have bioactive ligands in ChEMBL, and 626 (58.2%) can interact with electrophiles in SLCABPP (Figure 2a & Figure S2b). Inspired by Pharos, a druggable genome resource²⁹, we labeled E3 ligases into E3drug, E3chem, E3cova, and E3dark (see Methods). Among the identified E3 ligases, 127 (11.8%) were labeled as E3drug, 75 (7.0%) as E3chem, and 484 (45.0%) as E3cova. Additionally, we quantified the number of ligand sources associated with the E3 ligases. We identified that 55 (5.2%) E3 ligases can interact with all three sources of ligands (Figure 2a & Figure S2a), including the clinically used VHL and CRBN, and the experimentally explored KEAP1, XIAP, MDM2, BIRC2, and AHR. Beyond these E3 ligases, 48 have not yet been reported to be co-opted in PROTAC, pointing them as potential candidates for expanding E3 ligases for PROTACs. For instance, Tripartite-motif protein 24 (TRIM24)³⁰ has

evidence of available ligands from drug, ChEMBL, and SLCABPP (Figure 2b), example ligands including salicylaldehyde (ChEMBL108925) and the inhibitor (DGIdb: 252166607). In addition to the count of ligand sources, we also quantified the total number of ligands per E3 ligase. 77 (7.2%) E3 ligases have over 300 ligands, including 7 out of 12 co-opted E3 ligases (Figure 2c).”(page 7)

Reviewer #3 (Remarks to the Author):

This article presents an integration of comprehensive data from multiple resources to investigate the activity of E3 ligases in human cells as a key factor in target protein degradation. The study pipeline is well-established, and the data sources are comprehensive. Additionally, the authors have created a user-friendly web portal to search for the data and present the results of the study.

Response: We thank the reviewer for her/his positive evaluation of our work.

However, before publishing, there are some minor revisions that should be considered:

1. In the expression data, the authors should provide clarification on all pre-processing analyses, such as the removal of genes with very low expression, identification and removal of possible batch effects, and identification and removal of outlier samples.

Response: Thanks. Given the large volume of omics data involved, we have taken great care in selecting our data sources to ensure quality. We opted to utilize the UCSC Toil RNAseq Recompute Compendium for bulk transcriptomics expression data derived from TCGA and GTEx⁴⁵. This compendium processes both TCGA and GTEx using a unified pipeline and includes quality control measures and batch effect mitigation. We have implemented median expression level analysis to minimize outlier effects. For protein expression, we obtained data from the Human Protein Atlas pathology, and binarized samples into high-medium and low-NA groups, following the methodology of a recent study²³.

Regarding the tumor single-cell transcriptomics data, we chose the Tumor Immune Single-cell Hub (TISCH), a scRNA-seq database specializing in the tumor microenvironment (TME)⁴⁶, to examine the expression level of E3 ligases in tumors at the single-cell level. All single-cell datasets in TISCH underwent uniform quality control, clustering, and cell-type annotation. We only utilized those single-cell datasets that originate from humans, are untreated, and are analyzed via 10X Genomics. For single-cell transcriptomics data in normal tissues, we used Tabula Sapiens, a benchmark human cell atlas that provides a preliminary map of nearly 500,000 cells across 24 organs from 15 normal human subjects⁴⁷. Noteworthy features of Tabula Sapiens include rapid sample processing, expert cell annotation, and rigorous quality control. We clarified that the most pre-processing was conducted by the original data sources. For example, TISCH has quantified and minimized the batch effects across datasets⁴⁶.

We added this detailed information in the revised manuscript, in the Method section (page 22 -23).

2. The public PPI datasets contain a high rate of false positive interactions, such as those obtained solely through text mining. The authors should explain how they reduced the false positive rate in the PPI data.

Response: Thanks for the insightful suggestion. We indeed have performed further analysis to reduce the potential false positive rate. For PPI sources obtained through text mining (e.g., BioGrid, IntAct, Reactome), we collected details on interaction types and experimental systems (when available), which are annotated in our web portal. While we have retained all PPIs to offer a

comprehensive dataset, we've marked certain interactions with an asterisk. These marked PPIs, which were not based on physical association or were not identified via methods such as yeast two-hybrid studies, affinity purification-mass spectrometry, protein 3D structures, or low-throughput experiments, serve as indicators for users that there might be a risk of false positives.

Additionally, we have revised the Methods section related to PPIs to provide a clearer understanding of the quality control process implemented for collected PPIs. The pertaining text now reads: “General PPIs between E3 ligase and target protein were collected from BioGrid⁴⁸ (<https://thebiogrid.org/>), IntAct⁴⁹ (<https://www.ebi.ac.uk/intact/>), and Reactome⁵⁰ (<https://reactome.org/>), HuRI⁵¹ (<http://www.interactome-atlas.org/>), STRING⁵² (<https://string-db.org/>). To ensure the quality of the incorporated PPIs, we extracted PPIs from literature curation sources such as BioGrid, IntAct, and Reactome, carefully removing any non-human entries. We marked PPIs that were in physical association identified via methods such as yeast two-hybrid studies, affinity purification-mass spectrometry, protein 3D structures, or low-throughput experiments, serving as indicators for users that these PPIs were obtained in a high confidence. Entries from HuRI were retained due to their comprising solely of experimentally verified human binary protein interactions. PPIs from STRING were combined with known and predicted PPIs, and we only retained those human-related and with high confidence (score >700). PPIs specific for E3-substrate interactions (ESIs) were retrieved from from UbiBrowser2.0²⁹ (<http://ubibrowser.bio-it.cn/>), retaining only human ESIs documented in literature and high-confidence predicted ESIs with scores exceeding 0.7.”(page 24)

Target	Detection Methods	PPI type	Source	Interface E3	Interface Target	High Confidence*
ABTB2	Affinity Capture-MS	physical	BioGrid	-	-	Yes
ACTR6	Affinity Capture-MS	physical	BioGrid	-	-	Yes
AFF4	Affinity Capture-MS	physical	BioGrid	-	-	Yes
AHSG	Affinity Capture-MS	physical	BioGrid	-	-	Yes
ALDOA	Affinity Capture-MS	physical	BioGrid	-	-	Yes
ANXA5	Affinity Capture-MS	physical	BioGrid	-	-	Yes

3.It would be interesting to conduct an enrichment analysis on the E3 ligase targets to investigate the biological processes and pathways involved and their differences in normal and tumour samples.

Response: Thanks for this valuable suggestion. We agree that the enrichment analysis might uncover the diverse functions of the gene. For instance, we performed Gene Ontology (GO) and KEGG pathway enrichment analysis on potential targets of TRIM24. According to the UniProt gene function annotation (<https://www.uniprot.org/uniprotkb/O15164/entry>), TRIM24 functions include transcriptional control, E3 ligase activity mediating TP53 degradation, among others. Both GO and KEGG enrichment analyses underscore the role of transcription regulation, with the KEGG enrichment analysis indicating enrichment in multiple cancer pathways.

Figure. GO enrichment analysis of potential targets of TRIM24

Figure. KEGG enrichment analysis of potential targets of TRIM24

Another example pertains to the enrichment analysis conducted on proteins interacting with CRBN. As per the UniProt gene function annotation (<https://www.uniprot.org/uniprotkb/Q96SW2/entry>), CRBN's functions encompass E3 ligase activity, regulation of large-conductance calcium-activated potassium channels, among others. The pathways enriched in this context include those involved with transporters and calcium signaling pathways, among others.

Figure. GO enrichment analysis of potential targets of CRBN

Figure. KEGG enrichment analysis of potential targets of CRBN

As evident in these examples, the enriched pathways are highly diverse and closely tied to the endogenous functions of these proteins. Therefore, we have incorporated GO and KEGG

enrichment analysis into our web portal to cater to researchers' interests (see screenshot below). We added this in our revised manuscript, stating that “Recognizing that the PPI of E3 ligases might provide insights into the gene's inherent functions, our web portal incorporates both Gene Ontology and KEGG enrichment analyses on proteins interacting with each E3 ligase.” (page 16).

It's important to note that it is challenging to determine whether targets of E3 ligases are generally enriched in tumor or normal tissues, as this can significantly vary depending on the specific E3 ligase and the biological context, that is the tumor type. We assume that such enrichment analysis are out of scope of the current work and could be performed interpedently by readers according to their specific interests.

PPI

GO Enrichment KEGG Enrichment

Show entries Search:

Target	Detection Methods	PPI type	Source	Interface E3	Interface Target	High Confidence*
ABTB2	Affinity Capture-MS	physical	BioGrid	-	-	Yes
ACTR6	Affinity Capture-MS	physical	BioGrid	-	-	Yes
AFF4	Affinity Capture-MS	physical	BioGrid	-	-	Yes
AHSG	Affinity Capture-MS	physical	BioGrid	-	-	Yes

View enrichment analysis result

4. Investigating related pathways for E3 ligase targets or for extended PPIs (indirect targets) could be useful in identifying any possible side effects of specific protein degradation.

Response: We thank for the suggestion. We added targets enriched pathways in the data portal (please refer to above comment). A single E3 ligase may interact with hundreds/thousands of proteins, and each of these proteins could further interact with hundreds/thousands more. For example, in the high confidence STRING human dataset, 298 unique proteins interact with VHL, leading to a total of 98,259 unique extended PPIs. Given the current lack of empirical evidence on how to further reliably rank or filter these interactions, we find that this extensive network of extended PPIs would likely be too complex for readers to navigate and understand (or, for this matter, to ourselves, to be honest...). Therefore, the possible inclusion of extended PPI information should in our minds be best left for a future study.

Reference

1. Bondeson, D. P. et al. Lessons in PROTAC Design from Selective Degradation with a Promiscuous Warhead. *Cell Chemical Biology* 25, 78-87.e5 (2018).
2. Schneider, M. et al. The PROTACtable genome. *Nat Rev Drug Discov* (2021) doi:10.1038/s41573-021-00245-x.
3. Békés, M., Langley, D. R. & Crews, C. M. PROTAC targeted protein degraders: the past is prologue. *Nat Rev Drug Discov* 1–20 (2022) doi:10.1038/s41573-021-00371-6.
4. Alexandrov, L. B. et al. The repertoire of mutational signatures in human cancer. *Nature* 578, 94–101 (2020).
5. Calabrese, C. et al. Genomic basis for RNA alterations in cancer. *Nature* 578, 129–136 (2020).
6. Yuan, Y. et al. Comprehensive molecular characterization of mitochondrial genomes in human cancers. *Nat Genet* 52, 342–352 (2020).
7. Rodriguez-Martin, B. et al. Pan-cancer analysis of whole genomes identifies driver rearrangements promoted by LINE-1 retrotransposition. *Nat Genet* 52, 306–319 (2020).
8. Hoadley, K. A. et al. Cell-of-Origin Patterns Dominate the Molecular Classification of 10,000 Tumors from 33 Types of Cancer. *Cell* 173, 291-304.e6 (2018).
9. Chen, H. et al. A Pan-Cancer Analysis of Enhancer Expression in Nearly 9000 Patient Samples. *Cell* 173, 386-399.e12 (2018).
10. Chiu, H.-S. et al. Pan-Cancer Analysis of lncRNA Regulation Supports Their Targeting of Cancer Genes in Each Tumor Context. *Cell Reports* 23, 297-312.e12 (2018).
11. Berger, A. C. et al. A Comprehensive Pan-Cancer Molecular Study of Gynecologic and Breast Cancers. *Cancer Cell* 33, 690-705.e9 (2018).
12. Gong, J. et al. A Pan-cancer Analysis of the Expression and Clinical Relevance of Small Nucleolar RNAs in Human Cancer. *Cell Rep* 21, 1968–1981 (2017).
13. Ye, Y. et al. The Genomic Landscape and Pharmacogenomic Interactions of Clock Genes in Cancer Chronotherapy. *Cell Syst* 6, 314-328.e2 (2018).
14. Zhang, Z. et al. Transcriptional landscape and clinical utility of enhancer RNAs for eRNA-targeted therapy in cancer. *Nat Commun* 10, 4562 (2019).
15. Ye, Y. et al. Sex-associated molecular differences for cancer immunotherapy. *Nat Commun* 11, 1779 (2020).
16. Ye, Y. et al. Characterization of Hypoxia-associated Molecular Features to Aid Hypoxia-Targeted Therapy. *Nat Metab* 1, 431–444 (2019).
17. Zhao, Q., Zhao, H., Zheng, K. & Wang, J. HyperAttentionDTI: improving drug–protein interaction prediction by sequence-based deep learning with attention mechanism. *Bioinformatics* 38, 655–662 (2022).
18. Wishart, D. S. et al. DrugBank 5.0: a major update to the DrugBank database for 2018. *Nucleic Acids Research* 46, D1074–D1082 (2018).
19. He, M. et al. PROTACs: great opportunities for academia and industry (an update from 2020 to 2021). *Sig Transduct Target Ther* 7, 1–64 (2022).
20. Moreau, K. et al. Proteolysis-targeting chimeras in drug development: A safety perspective. *Br J Pharmacol* 177, 1709–1718 (2020).
21. Chen, C. et al. Recent Advances in Pro-PROTAC Development to Address On-Target Off-Tumor Toxicity. *J. Med. Chem.* 66, 8428–8440 (2023).
22. Khan, S. et al. A selective BCL-XL PROTAC degrader achieves safe and potent antitumor activity. *Nat Med* 25, 1938–1947 (2019).

23. MacKay, M. et al. The therapeutic landscape for cells engineered with chimeric antigen receptors. *Nat Biotechnol* 38, 233–244 (2020).
24. Jing, Y. et al. Expression of chimeric antigen receptor therapy targets detected by single-cell sequencing of normal cells may contribute to off-tumor toxicity. *Cancer Cell* 39, 1558–1559 (2021).
25. Jing, Y. et al. Association Between Sex and Immune-Related Adverse Events During Immune Checkpoint Inhibitor Therapy. *JNCI: Journal of the National Cancer Institute* djab035 (2021) doi:10.1093/jnci/djab035.
26. Jing, Y., Yang, J., Johnson, D. B., Moslehi, J. J. & Han, L. Harnessing big data to characterize immune-related adverse events. *Nat Rev Clin Oncol* 19, 269–280 (2022).
27. Ge, Z. et al. Integrated Genomic Analysis of the Ubiquitin Pathway across Cancer Types. *Cell Reports* 23, 213–226.e3 (2018).
28. Liu, L. et al. UbiHub: a data hub for the explorers of ubiquitination pathways. *Bioinformatics* 35, 2882–2884 (2019).
29. Wang, X. et al. UbiBrowser 2.0: a comprehensive resource for proteome-wide known and predicted ubiquitin ligase/deubiquitinase–substrate interactions in eukaryotic species. *Nucleic Acids Research* (2021) doi:10.1093/nar/gkab962.
30. Ge, K., X, Z. & Ze, F. PPAR- γ AF-2 domain functions as a component of a ubiquitin-dependent degradation signal. *Obesity (Silver Spring, Md.)* 17, (2009).
31. Hou, Y. et al. PPAR γ E3 ubiquitin ligase regulates MUC1-C oncoprotein stability. *Oncogene* 33, 5619–5625 (2014).
32. Lee, J. H. et al. Degradation of selenoprotein S and selenoprotein K through PPAR γ -mediated ubiquitination is required for adipocyte differentiation. *Cell Death Differ* 26, 1007–1023 (2019).
33. Hou, Y., Moreau, F. & Chadee, K. PPAR γ is an E3 ligase that induces the degradation of NF κ B/p65. *Nat Commun* 3, 1300 (2012).
34. Briassouli, P., Chan, F. & Linardopoulos, S. The N-terminal domain of the Aurora-A Phe-31 variant encodes an E3 ubiquitin ligase and mediates ubiquitination of IkappaBalpha. *Hum Mol Genet* 15, 3343–3350 (2006).
35. Yang, C. et al. Effects of AURKA-mediated degradation of SOD2 on mitochondrial dysfunction and cartilage homeostasis in osteoarthritis. *J Cell Physiol* 234, 17727–17738 (2019).
36. Dale, B. et al. Advancing targeted protein degradation for cancer therapy. *Nat Rev Cancer* (2021) doi:10.1038/s41568-021-00365-x.
37. Hines, J., Lartigue, S., Dong, H., Qian, Y. & Crews, C. M. MDM2-Recruiting PROTAC Offers Superior, Synergistic Antiproliferative Activity via Simultaneous Degradation of BRD4 and Stabilization of p53. *Cancer Research* 79, 251–262 (2019).
38. Marcellino, B. et al. Development of an MDM2 Degradator for Treatment of Acute Leukemias. *Blood* 138, 1866 (2021).
39. Ohoka, N. et al. Development of Small Molecule Chimeras That Recruit AhR E3 Ligase to Target Proteins. *ACS Chem. Biol.* 14, 2822–2832 (2019).
40. Zhang, X. et al. Discovery of IAP-recruiting BCL-XL PROTACs as potent degraders across multiple cancer cell lines. *European Journal of Medicinal Chemistry* 199, 112397 (2020).
41. Yoshida, H. et al. Elucidation of the aberrant 3' splice site selection by cancer-associated mutations on the U2AF1. *Nat Commun* 11, 4744 (2020).
42. Cheruiyot, A. et al. Nonsense-Mediated RNA Decay Is a Unique Vulnerability of Cancer Cells Harboring SF3B1 or U2AF1 Mutations. *Cancer Research* 81, 4499–4513 (2021).

43. Nguyen, D.-T. et al. Pharos: Collating protein information to shed light on the druggable genome. *Nucleic Acids Research* 45, D995–D1002 (2017).
44. Schapira, M., Calabrese, M. F., Bullock, A. N. & Crews, C. M. Targeted protein degradation: expanding the toolbox. *Nat Rev Drug Discov* 18, 949–963 (2019).
45. Vivian, J. et al. Toil enables reproducible, open source, big biomedical data analyses. *Nat Biotechnol* 35, 314–316 (2017).
46. Sun, D. et al. TISCH: a comprehensive web resource enabling interactive single-cell transcriptome visualization of tumor microenvironment. *Nucleic Acids Research* 49, D1420–D1430 (2021).
47. Consortium, T. T. S. & Quake, S. R. The Tabula Sapiens: a multiple organ single cell transcriptomic atlas of humans. 2021.07.19.452956 Preprint at <https://doi.org/10.1101/2021.07.19.452956> (2021).
48. Oughtred, R. et al. The BioGRID interaction database: 2019 update. *Nucleic Acids Research* 47, D529–D541 (2019).
49. Orchard, S. et al. The MIntAct project—IntAct as a common curation platform for 11 molecular interaction databases. *Nucleic Acids Research* 42, D358–D363 (2014).
50. Gillespie, M. et al. The reactome pathway knowledgebase 2022. *Nucleic Acids Research* 50, D687–D692 (2022).
51. Luck, K. et al. A reference map of the human binary protein interactome. *Nature* 580, 402–408 (2020).
52. Szklarczyk, D. et al. STRING v10: protein-protein interaction networks, integrated over the tree of life. *Nucleic Acids Res* 43, D447–452 (2015).

Reviewers' Comments:

Reviewer #1:

Remarks to the Author:

The authors have made rather minimal changes to their manuscript which do not in my view substantially address the concerns that I and other reviewers raised in our initial reports. In particular, my concern regarding the incorporation of results from an unvalidated predictive model still stands.

My views on this paper are therefore unchanged from the original version; whilst this is a useful resource that could be published in a specialised data/databases journal it does not seem to me that this paper reaches the level expected for Nature Comms.

Reviewer #2:

Remarks to the Author:

Authors have addressed all my concerns very satisfactorily. I would like to congratulate them for their works and the thorough revision.

Reviewer #3:

Remarks to the Author:

The authors have responded to all of my concerns and I think the manuscript can be published in the journal.

Re: NCOMMS-23-12374A

Reviewer #1 (Remarks to the Author):

The authors have made rather minimal changes to their manuscript which do not in my view substantially address the concerns that I and other reviewers raised in our initial reports. In particular, my concern regarding the incorporation of results from an unvalidated predictive model still stands.

My views on this paper are therefore unchanged from the original version; whilst this is a useful resource that could be published in a specialised data/databases journal it does not seem to me that this paper reaches the level expected for Nature Comms.

Response: We made substantial revision according to the suggestions from the reviewers, including data presentation, audience scope discussion, downloading request. Regarding the prediction results, we revised our approach based on the HyperAttentionDTI study's format¹, emphasizing interactions with the highest prediction probabilities. In our updated manuscript, only the top drug-target interactions for each E3 ligase are showcased, with the default being the top 10. Users can adjust this to view anywhere from 1 to 500 interactions and can also set a specific threshold to filter predictions based on a desired cutoff.

We have adequately addressed other reviewers' comments, as indicated by very positive comments from other reviewers. We also carefully addressed reviewer 1's original comments.

Regarding to the level expected for Nature Comms, we believe it is an arbitrary argument, and the editor will have a better judgement on this. Recent studies demonstrated the power of rigorous analysis from large-scale data. For example, the recent TCGA/ICGC Pan-cancer project published a series of high-profile papers in Nature and Cell series journals²⁻⁹. These comprehensive analyses provide insights for the research community for further investigations. Our group also significantly contributed to the cancer genomics field¹⁰⁻¹⁴. Please also note, the PROTACTable Genome analysis also utilized similar strategy without any experimental validation, and published in the top journal, Nature Reviews Drug Discovery¹⁵.

Reviewer #2 (Remarks to the Author):

Authors have addressed all my concerns very satisfactorily. I would like to congratulate them for their works and the thorough revision.

Response: We thank the reviewer for the positive evaluation of our revision.

Reviewer #3 (Remarks to the Author):

The authors have responded to all of my concerns and I think the manuscript can be published in the journal.

Response: We thank the reviewer for the positive evaluation of our revision.

REFERENCES

1. Zhao, Q., Zhao, H., Zheng, K. & Wang, J. HyperAttentionDTI: improving drug–protein interaction prediction by sequence-based deep learning with attention mechanism. *Bioinformatics* **38**, 655–662 (2022).
2. Alexandrov, L. B. *et al.* The repertoire of mutational signatures in human cancer. *Nature* **578**, 94–101 (2020).
3. Calabrese, C. *et al.* Genomic basis for RNA alterations in cancer. *Nature* **578**, 129–136 (2020).
4. Yuan, Y. *et al.* Comprehensive molecular characterization of mitochondrial genomes in human cancers. *Nat Genet* **52**, 342–352 (2020).
5. Rodriguez-Martin, B. *et al.* Pan-cancer analysis of whole genomes identifies driver rearrangements promoted by LINE-1 retrotransposition. *Nat Genet* **52**, 306–319 (2020).
6. Hoadley, K. A. *et al.* Cell-of-Origin Patterns Dominate the Molecular Classification of 10,000 Tumors from 33 Types of Cancer. *Cell* **173**, 291-304.e6 (2018).
7. Chen, H. *et al.* A Pan-Cancer Analysis of Enhancer Expression in Nearly 9000 Patient Samples. *Cell* **173**, 386-399.e12 (2018).
8. Chiu, H.-S. *et al.* Pan-Cancer Analysis of lncRNA Regulation Supports Their Targeting of Cancer Genes in Each Tumor Context. *Cell Reports* **23**, 297-312.e12 (2018).
9. Berger, A. C. *et al.* A Comprehensive Pan-Cancer Molecular Study of Gynecologic and Breast Cancers. *Cancer Cell* **33**, 690-705.e9 (2018).
10. Gong, J. *et al.* A Pan-cancer Analysis of the Expression and Clinical Relevance of Small Nucleolar RNAs in Human Cancer. *Cell Rep* **21**, 1968–1981 (2017).
11. Ye, Y. *et al.* The Genomic Landscape and Pharmacogenomic Interactions of Clock Genes in Cancer Chronotherapy. *Cell Syst* **6**, 314-328.e2 (2018).
12. Zhang, Z. *et al.* Transcriptional landscape and clinical utility of enhancer RNAs for eRNA-targeted therapy in cancer. *Nat Commun* **10**, 4562 (2019).
13. Ye, Y. *et al.* Sex-associated molecular differences for cancer immunotherapy. *Nat Commun* **11**, 1779 (2020).
14. Ye, Y. *et al.* Characterization of Hypoxia-associated Molecular Features to Aid Hypoxia-Targeted Therapy. *Nat Metab* **1**, 431–444 (2019).
15. Schneider, M. *et al.* The PROTACtable genome. *Nat Rev Drug Discov* (2021) doi:10.1038/s41573-021-00245-x.